# Enhanced genetic fine mapping accuracy with Bayesian Linear Regression models in diverse genetic architectures

Merina Shrestha[1]*, Zhonghao Bai[1], Tahereh Gholipourshahraki[1], Astrid J. Hjelholt[2,3,4], Sile Hu[5], Mads Kjolby[2,3,4], Palle Duun Rohde[6], Peter Sørensen[1]*

1 Center for Quantitative Genetics and Genomics, Aarhus University, Aarhus, Denmark, 2 Department of Biomedicine, Aarhus University, Aarhus, Denmark, 3 Department of Clinical Pharmacology, Aarhus University Hospital, Aarhus, Denmark, 4 Steno Diabetes Center Aarhus, Aarhus University Hospital, Aarhus, Denmark, 5 Human Genetic Center of Excellence, Novo Nordisk Research Centre Oxford, Oxford, United Kingdom, 6 Genomic Medicine, Department of Health Science and Technology, Aalborg University, Aalborg, Denmark

* merina.shrestha@sund.ku.dk (MS); pso@qgg.au.dk (PS)

## Abstract

We evaluated Bayesian Linear Regression (BLR) models with BayesC and BayesR priors as statistical genetic fine-mapping tools, comparing their performance to established methods such as FINEMAP and SuSiE. Through extensive simulations and analyses of UK Biobank (UKB) phenotypes, we assessed F1 classification scores and predictive accuracy across models. Simulations encompassed diverse genetic architectures varying in polygenicity, heritability, causal SNP proportions, and disease prevalence. In the empirical analyses, we used over 6.6 million imputed SNPs and phenotypic data from more than 335,000 UKB participants. Our results show that BLR models, particularly those using the BayesR prior, consistently achieved higher F1 scores than the external methods, but having comparable predictive accuracy. Applying the BLR model at the region-wide level generally yielded better F1 scores than the genome-wide approach, except for traits with high polygenicity. These findings highlight BLR models as accurate and robust tools for statistical fine mapping in both simulated and empirical genetic datasets.

## Author summary

Understanding which genetic variants influence complex traits and diseases is essential for gaining biological insight and developing new treatments. However, large-scale genetic studies often identify many potential genetic variants, making it difficult to determine which ones are the causal genetic variants. This challenge arises in part because nearby variants in the genome tend to be inherited together, which complicates statistical inference. To alleviate this problem,

which permits unrestricted use, distribution, and reproduction in any medium, provided the original author and source are credited.

**Data availability statement:** S2–S5 Tables contain the processed fine-mapping results for all methods across all simulations and can be used to reproduce the figures and results presented in this revised manuscript. Example scripts for the fine-mapping procedures involving the evaluated BLR models are available at: https://psoerensen.github.io/gact/Document/Finemapping_bayesian_linear_regression_simulated_data.html. Functions for simulating the different scenarios used in this study, performing fine-mapping, and generating credible sets are implemented in the R package qgg, which is available at https://psoerensen.github.io/qgg/. The genotype and phenotype data used in our analyses are available from UK Biobank (https://www.ukbiobank.ac.uk/).

**Funding:** MK, PDR, and PS obtained funding from Novo Nordisk Foundation through the drug discovery platform, Open Discovery Innovation Network (ODIN) under grant number "NNF20SA0061466". This funding aims to foster collaboration between universities and companies promoting long-term benefits of innovation. The funders had no role in study design, data collection and analysis, decision to publish, or preparation of the manuscript.

**Competing interests:** The authors have declared that no competing interests exist.

statistical fine mapping, a technique that narrows down the list of genetic variants to the most likely causal ones, is often used. In this study, we propose using Bayesian Linear Regression (BLR) models for statistical genetic fine mapping. By incorporating assumptions about how many genetic variants influence a trait and how large their effects are, called prior distributions, these models increase the statistical power to detect causal variants. We tested the BLR models using both simulated data and empirical data from UK Biobank (UKB), which includes genetic and health information from hundreds of thousands of people. Our results from the simulation studies show that BLR models perform comparably to state-of-the-art fine-mapping methods in identifying causal genetic variants. This performance was evaluated by F1 classification score and area under the receiver operating characteristic curve (AUC). When applied to phenotypes from UKB, BLR models achieve predictive accuracy comparable to state-of-the-art fine mapping tools. These findings suggest that BLR models are a powerful and flexible tool identifying genetic variants that contribute to complex traits and diseases.

## Introduction

To better understand the genetic architecture of complex traits and multifactorial diseases, it is essential to identify the genetic variants that are most likely causal or in linkage disequilibrium (LD) with the causal variants. Genome-wide association studies (GWAS) often identify numerous associated variants due to long-range LD, which complicates statistical inference [1]. Therefore, post-GWAS statistical fine-mapping analyses are typically required to refine these signals and pinpoint the variants most likely responsible for the phenotype [2]. This task is critical, as it informs follow-up studies such as large-scale replication or functional assays aimed at gaining biological insight and informing potential clinical applications, including drug discovery and repurposing [3].

Fine-mapping methods generally assume that the causal variant(s) are present in the dataset [4]. With growing evidence for the presence of multiple causal variants within a locus, Bayesian fine-mapping methods have been developed to model this complexity [5]. These approaches estimate the posterior inclusion probability (PIP) for each variant, the probability that a variant has a non-zero effect, thereby quantifying the likelihood that a variant is causal [2]. In addition, Bayesian methods can incorporate prior knowledge about the genetic architecture of a trait, including assumptions about the number, frequency, and effect sizes of causal variants, which helps improve statistical power [6,7].

Several methods have been proposed for fine mapping based on different modeling assumptions. For example, FINEMAP [8] uses GWAS summary statistics and a Shotgun Stochastic Search (SSS) algorithm to explore likely causal configurations. In contrast, SuSiE [9] models sparse signals by summing multiple single-effect components using an iterative Bayesian stepwise selection approach. SuSiE has been

extended to work with summary statistics (SuSiE-RSS) [10]. Recently, SuSiE-Inf and FINEMAP-Inf [6] were developed to model both sparse and infinitesimal genetic effects, marginalizing over the residuals and LD structure within loci.

In this context, we investigated the use of Bayesian Linear Regression (BLR) models for fine mapping. BLR models have been widely applied in genetic prediction, estimation of genetic parameters, and studying genetic architecture [11]. These models jointly estimate marker effects while accounting for LD and allow flexible modeling of effect size distributions through the choice of priors. Specifically, we evaluated two commonly used priors: BayesC, which assumes a fixed proportion of variants have zero effect [12], and BayesR, which extends this by assigning variants to multiple effect size categories, enabling both variable selection and effect size shrinkage [13].

While BLR models have been extensively used for polygenic prediction [14–16], their potential as fine-mapping tools has received limited attention. Few studies have systematically evaluated their ability to identify causal variants using metrics like precision, recall, and F1 score [6,10]. To address this gap, we assessed BLR models in both region-level and genome-wide implementations, comparing how prior assumptions and model scope influence fine-mapping resolution and accuracy.

In fine mapping, due to complex LD patterns, individual causal variants are often not distinguishable. Therefore, credible sets of likely causal variants are prioritized [4]. These sets consist of the smallest group of variants whose cumulative PIP exceeds a certain threshold (e.g., 0.9 or 0.99), ensuring that they likely contain the true causal variant(s) [5,17]. A key goal is to make these sets as small as possible while maintaining high confidence, thereby aiding interpretation and downstream validation.

In this study, we evaluated the efficiency of BLR models with BayesC and BayesR priors as fine-mapping tools using GWAS summary statistics. Through simulations, we constructed credible sets and assessed model performance using precision, recall, and F1 score. We also evaluated credible set properties and prediction accuracy across five binary and five quantitative phenotypes from the UK Biobank [18]. Results from BLR models were compared to state-of-the-art fine-mapping methods including FINEMAP, SuSiE-RSS, SuSiE-Inf, and FINEMAP-Inf. Finally, we validated the BLR model by applying it to the fine mapping of type 2 diabetes (T2D) within the UK Biobank dataset.

## Materials and methods

### UKB genetic data

UKB genotyped genetic variants were used for the simulation study whereas imputed genetic variants were used in analysis of UKB phenotypes. To obtain a genetic homogeneous study population we restricted our analyses to unrelated British Caucasians and excluded individuals with more than 5,000 missing variants or individuals with autosomal aneuploidy. Remaining ($n = 335{,}532$) White British unrelated individuals (WBU) were used for analyses. Genetic variants with minor allele frequency <0.01, call rate <0.95 and the variants deviating from Hardy-Weinberg equilibrium ($P$-value $< 1 \times 10^{-12}$) were excluded. Further, we excluded genetic variants located within the major histocompatibility complex (MHC), having ambiguous allele (i.e., GC or AT), were multi-allelic or an indel [19]. This resulted in a total of 533,679 single nucleotide polymorphisms (SNPs).

For the imputed data, firstly genetic variants with genotype probability of 70% (hard-call threshold 0.7) were converted to genotypes followed by retaining variants with imputation score >= 0.8 using PLINK 2.0 [20]. The same quality control criteria were applied to the imputed genetic variants as for the genotyped data, except that we included MHC in the UKB phenotypes as this region contains many known disease-associated variants. After quality control of UKB imputed genetic data 6,627,732 SNPs were left for analyses.

### Simulation of quantitative and binary phenotypes

**Quantitative phenotypes.** To simulate genetic architectures from low to high polygenicity, we simulated quantitative phenotypes with heritability ($h^2_{SNP}$) of 30% and 10%, with two different proportions of causal SNPs (π), 0.1% and 1%, chosen randomly from the observed genotypes.

To reflect different underlying genetic architectures of the simulated phenotypes ($h^2_{SNP} = (0.1; 0.3)$ and $\pi = (0.0001; 0.01)$) two types of genetic architectures were simulated (genetic architecture 1 and 2, $GA_1$ and $GA_2$). In $GA_1$, $m_C$ causal SNPs effects (**b**) were sampled from a normal distribution (with mean of 0 and variance given by $\sigma^2_g/m_C$) (Eq. 1):

$$\mathbf{y} = \sum_{i=1}^{m_C} \mathbf{w}_i b_i + \mathbf{e},$$
(1)

where **y** is the phenotype for, $b_i$ is the estimate of the $i$-th SNP effect. The $Var(y) = 1$ such that $\sigma^2_g$ is equal to $h^2_{SNP}$. The residual, **e**, has a normal distribution with mean=0 and variance = $\sigma^2_g \times \left(1/\left(h^2_{snp}\right) - 1\right)$. The vector $\mathbf{w}_i$ represents the $i$-th centered and scaled genotype (Eq. 2),

$$\mathbf{w}_i = \frac{\mathbf{x}_i - 2p_i}{\sqrt{2p_i\left(1 - p_i\right)}}$$
(2)

where, $\mathbf{x}_i$ is the effect allele count at the $i$-th SNP, $p_i$ is the allele frequency of the $i$-th SNP.

For genetic architecture 2 ($GA_2$), the effects of causal SNPs were sampled from a mixture of three normal distributions such that 93% of the causal SNPs would have small effect sizes and the remaining 5% and 2% of the causal SNPs would have moderate and large effect sizes respectively. The proportion of causal genetic variants in each class was designed in a similar way as by Lloyd-Jones *et al.* [14]. Eq. 3:

$$\mathbf{y} = \sum_{i=1}^{m_{C_1}} \mathbf{w}_i b_i + \sum_{j=1}^{m_{C_2}} \mathbf{w}_j b_j + \sum_{k=1}^{m_{C_3}} \mathbf{w}_k b_k + \mathbf{e},$$
(3)

where, $b_i$, $b_j$, and $b_k$ are the effect of causal SNPs sampled from normal distribution with mean=0 and variance = $\left(0.6\sigma^2_g\right)/\left(0.93m_{C_1}\right)$, $\left(0.2\sigma^2_g\right)/\left(0.05m_{C_2}\right)$, and $\left(0.2\sigma^2_g\right)/\left(0.02m_{C_3}\right)$, respectively [14].

For each simulation scenario (eight in total=two levels of $h^2_{SNP}$, two values of π, and two genetic architectures), ten replicates were simulated. For each replicate within a given scenario, causal SNPs were randomly sampled without replacement. As a result, the likelihood of the same SNP being selected multiple times is extremely low. Even in the rare instance where this occurred, the simulated SNP effects would differ across replicates. The total sample of 335,532 were divided into ten replicates. Each replicate contained 80% of the randomly sampled data from the total samples.

**Two causal genetic variants.** To evaluate the performance of fine-mapping models in the presence of two causal SNPs within a fine-mapped region, we simulated quantitative phenotypes using 89,458 quality-controlled SNPs from chromosome 22, based on imputed genotype data from the UK Biobank. A comparative evaluation of fine-mapping methods was carried out using simulation studies involving two causal variants, across varying sample sizes (200K, 250K, and 300K) and effect size configurations. The three effect configurations were defined as follows: LL, with two large-effect causal variants ($b_i$ = 0.05); LS, with one large ($b_i$ = 0.05) and one small ($b_i$ = 0.01) effect variant; and SS, with two small-effect causal variants ($b_i$ = 0.01). Causal variants were selected by randomly choosing an index SNP on chromosome 22 and then selecting 1,000 SNPs upstream and 1,000 SNPs downstream, resulting in a region of 2,001 SNPs. Two SNPs within this region were randomly assigned as causal. Phenotypes were simulated with residual variance fixed at 1.0. Genotypes were centered and scaled, resulting in per-SNP heritabilities of 0.0025 for large effects and 0.001 for small effects. A total of 100 replicates were generated for each combination of effect configuration and sample size.

**Binary phenotypes.** To simulate the binary phenotypes, disease prevalence (*PV*) was introduced as simulation parameter. Two different *PV* of 5% and 15% were used. We simulated binary phenotypes from quantitative phenotypes. To simulate a binary phenotype with *PV* 5%, we chose top 5% of individuals with highest simulated quantitative values as cases and the remaining as controls for the total sample in a replicate. Each scenario of a quantitative phenotype gave rise to two different scenarios for binary phenotype, thus, resulting in 16 scenarios in total. Details on the different

scenarios for the quantitative and the binary phenotypes are presented S1 Table. The flowchart of design of the simulations is presented in Fig 1.

### Quantitative and binary phenotypes in UKB

For real data analysis five quantitative phenotypes and five binary phenotypes were selected for analysis (Table 1). The quantitative phenotypes were identified using specific field codes in the UKB data (see UKB showcase, Table 1). For all UKB phenotypes and covariates we used the first reported instance. We estimated the ratio of the waist circumference to the hip circumference (WHR). To define individuals as disease cases we used ICD10-codes from the data field "Diagnosis-main ICD10" along with codes from the self-reported information (Table 2). Individuals without the ICD10 diagnosis code for a particular disease were used as controls. Additional information, such as age at recruitment (p21022), biological sex at birth (p31), and the UKB assessment center (p54), was collected with the phenotypes. Detailed information regarding the number of samples, prevalence for the phenotypes is given in Tables 1 and 2.

For the subsequent analyses of the UKB phenotypes, the WBU UKB cohort was divided into five replicates of training (80%) and validation (20%).

### Genome-wide association study

For the eight different simulated quantitative phenotypes, each with ten independent replicates, we conducted GWAS using the R package qgg [21,22]. A single SNP linear regression model was applied without including any covariates, as no covariates were simulated. We applied the same regression model without covariates for the quantitative phenotypes designed under "Two causal genetic variants". For the 16 different simulated binary phenotypes each with ten independent replicates the GWAS was conducted using PLINK 1.9 [23] modelled as a logistic regression, again without adjustment for confounding effects.

For the real trait analyses of the UKB phenotypes, the GWAS were performed within the five replicates of training cohort. For T2D, the GWAS was also performed using the entire WBU UKB cohort to understand the biological mechanism underlying T2D (Fig 2). For the quantitative phenotypes we performed single SNP linear regression using the R package qgg [21,22], and for the binary phenotypes we used logistic regression within PLINK 1.9 [20]. We used top ten principal components (PCs) along with age, sex and the UKB assessment center as covariates in all GWAS of real traits. We computed PCs for WBU from 100K randomly sampled SNPs from the genotyped data after removing SNPs in the autosomal long-range LD regions [24] with pairwise correlation ($r^2$)>0.1 in 500Kb region, using PLINK 2.0 [20].

### The statistical model for fine mapping

The BLR model can be written as a multiple linear regression model, Eq. 4,

$$y = Xb + e,$$

(4)

where $y$ is a vector phenotypic values, $X$ is a matrix of centered and scaled genetic variants $x_i = (M_i - 2p_i)/\sqrt{2p_i(1-p_i)}$, with $M_i$ being the number of copies of the effect allele (i.e., 0, 1 or 2) for the $i$-th variant and $p_i$ is the allele frequency of the effect allele. The vector $b$ is the estimated SNP effects for each genetic variant, and $e$, the residual, are a priori assumed to be independently and identically distributed multivariate normal with null mean and covariance matrix $I\sigma_e^2$, where $I$ is the identity matrix.

The key parameter of interest in the multiple regression model is the SNP effects, $b$, which can be obtained by solving Eq. 5:

$$b = \left(X'X + I\frac{\sigma_e^2}{\sigma_b^2}\right)^{-1} X'y,$$

(5)

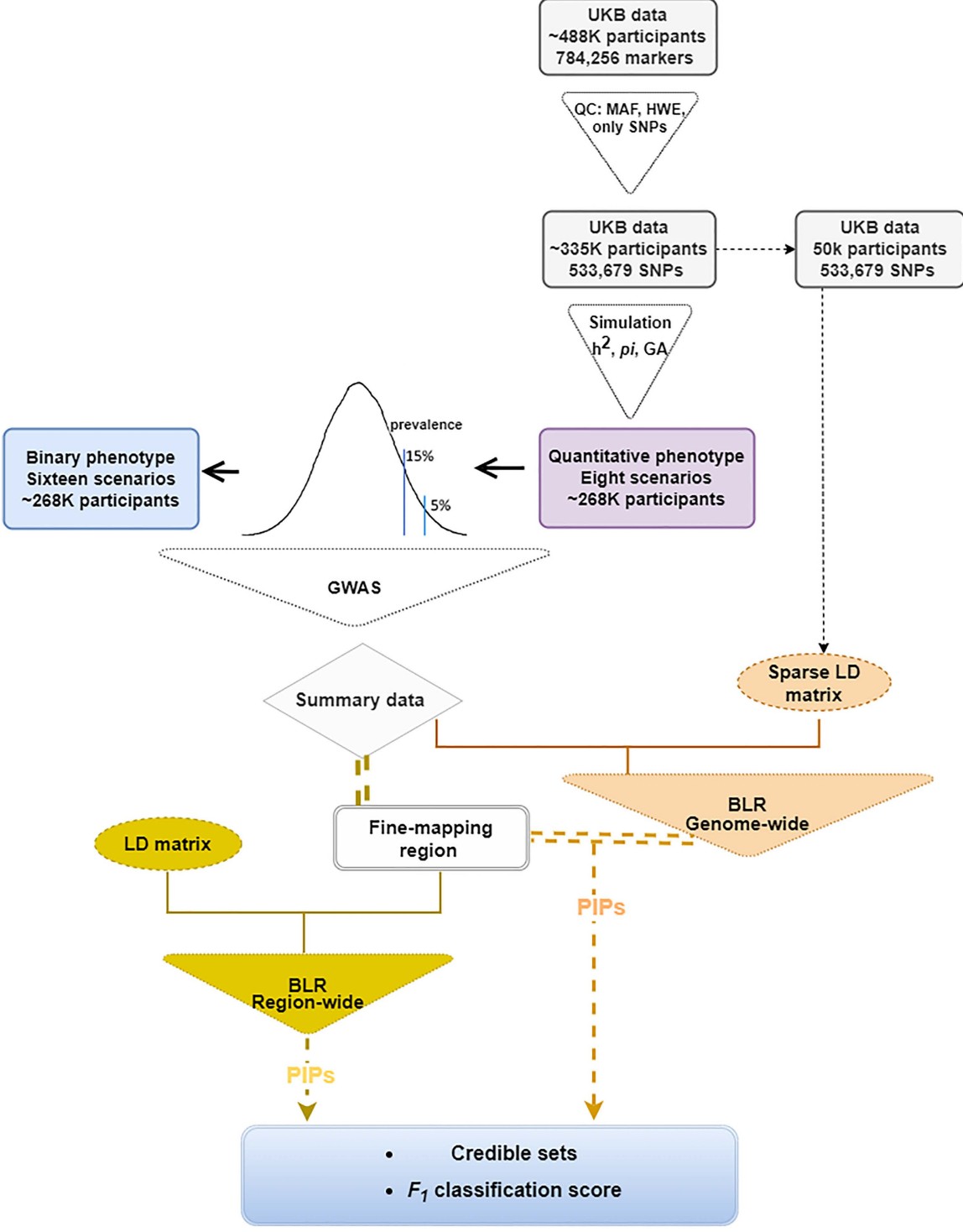

**Fig 1. Flowchart illustrating the design of the simulation scenarios for both quantitative and the phenotypes, followed by fine mapping using Bayesian Linear Regression models.** The BLR models were implemented in different ways, and the resulting posterior inclusion probability (PIPs) for SNPs were used to estimate the $F_1$ classification score based on the credible sets.

**Table 1. Details of the data fields along with the total number of non-missing samples, age (mean and standard deviation), number (No) of females and average value for the quantitative phenotypes with standard deviation (sd) represented in brackets.**

| UKB quantitative phenotypes | UKB data field | Total | Age [sd] | No females | Mean [sd] |
|---|---|---|---|---|---|
| Body Mass Index (BMI) | p21001 | 334,464 | 56.87 [7.98] | 179,309 | 27.4 [4.76] |
| Hip Circumference (HC) | p49 | 334,949 | 56.87 [7.98] | 179,517 | 103.44 [9.15] |
| Standing Height (Height) | p50 | 334,828 | 56.87 [7.98] | 179,492 | 168.86 [9.25] |
| Waist Circumference (WC) | p48 | 334,983 | 56.87 [7.98] | 179,532 | 90.37 [13.49] |
| Waist-Hip Ratio (WHR) | NA* | 334,917 | 56.87 [7.98] | 179,503 | 0.87 [0.09] |

\* The phenotype was calculated based on WC and HC.

**Table 2. Details on cases definitions for the UKB binary phenotypes based on ICD10 codes and self-reported diseases, total number of cases, controls along with the distribution of age (mean and standard deviation [sd]) and number (No) of females within cases and controls.**

| UKB Binary phenotypes | Definition of cases | | Cases | Controls | Age [sd] | | No females | |
|---|---|---|---|---|---|---|---|---|
| | ICD10 code | Self-reported code | | | Cases | Controls | Cases | Controls |
| Coronary Artery Disease (CAD) | I21; I22; I23; I24; I25 | 1075 | 34,726 | 300,806 | 61.24 [6.38] | 56.37 [7.99] | 10,845 | 168,989 |
| Hypertension (HTN) | I10 | 1065 | 129,580 | 205,952 | 59.75 [6.98] | 55.07 [8.04] | 60,859 | 118,975 |
| Psoriasis (PSO) | L40 | 1453 | 6628 | 328,904 | 57.15 [7.88] | 56.87 [7.98] | 3090 | 176,744 |
| Rheumatoid Arthritis (RA) | M06 | 1464 | 7955 | 327,577 | 59.6 [7.04] | 56.81 [7.99] | 5251 | 174,583 |
| Type 2 Diabetes (T2D) | E11 | 1220;1223 | 25,828 | 309,704 | 60.11 [6.9] | 56.6 [8.01] | 10,072 | 169,762 |

where, $\sigma_e^2$ is residual variance and $\sigma_b^2$ is the SNP-effect variance.

To solve this equation system, individual level data (genotypes [$X$] and phenotypes [$y$]) are required. If these are not available, it is possible to reconstruct $X'y$ and $X'X$ from a LD correlation matrix $B$ (from a population matched LD reference panel) and GWAS summary data [14] Eq. 6:

$$X'X = D^{0.5}BD^{0.5},$$
$$X'y = D\widetilde{b}$$

(6)

where $D_i = \frac{1}{\sigma_{b_i}^2 + \tilde{b}_i^2/n_i}$ if the genetic variants have been centered to mean 0, else $D_i = n_i$ if the genetic variants has been centered to mean 0 and scaled to variance of 1; thus, the diagonal elements of $D$ represent the per variant sample size. The vector $\widetilde{b}$ is marginal SNP effects obtained from a standard GWAS, with $\sigma_{b_i}^2$ being the variance of the marginal effects from GWAS. The LD correlation matrix, $B$, can be obtained as the squared Pearson's correlation ($r^2$) among genetic variants.

**Estimation of parameters using BLR models.** BLR models use an iterative algorithm based on Markov Chain Monte Carlo (MCMC) techniques, specifically Gibbs sampling, to estimate joint SNP effects. These estimates depend on several model parameters, including the probability of a variant being causal being causal ($\pi$), an overall SNP effect variance ($\sigma_b^2$), and the residual variance ($\sigma_e^2$). The posterior of the model parameters ($b,\sigma_b^2,\sigma_e^2$) depends on the likelihood of the observed data given the parameters, as well as prior distributions for those parameters, as described by Rohde et al. [21]. The posterior inclusion probability (PIP) of a SNP is defined as the proportion of MCMC iterations during which the SNP is included in the model with a non-zero effect [2,21].

The choice of prior for SNP effects should ideally reflect the genetic architecture of the phenotype. Most complex traits and diseases are polygenic, involving many causal variants with typically small effects [25]. Therefore, the effect prior should accommodate many small effects and a few larger ones. Although SNP effects are assumed to be uncorrelated a

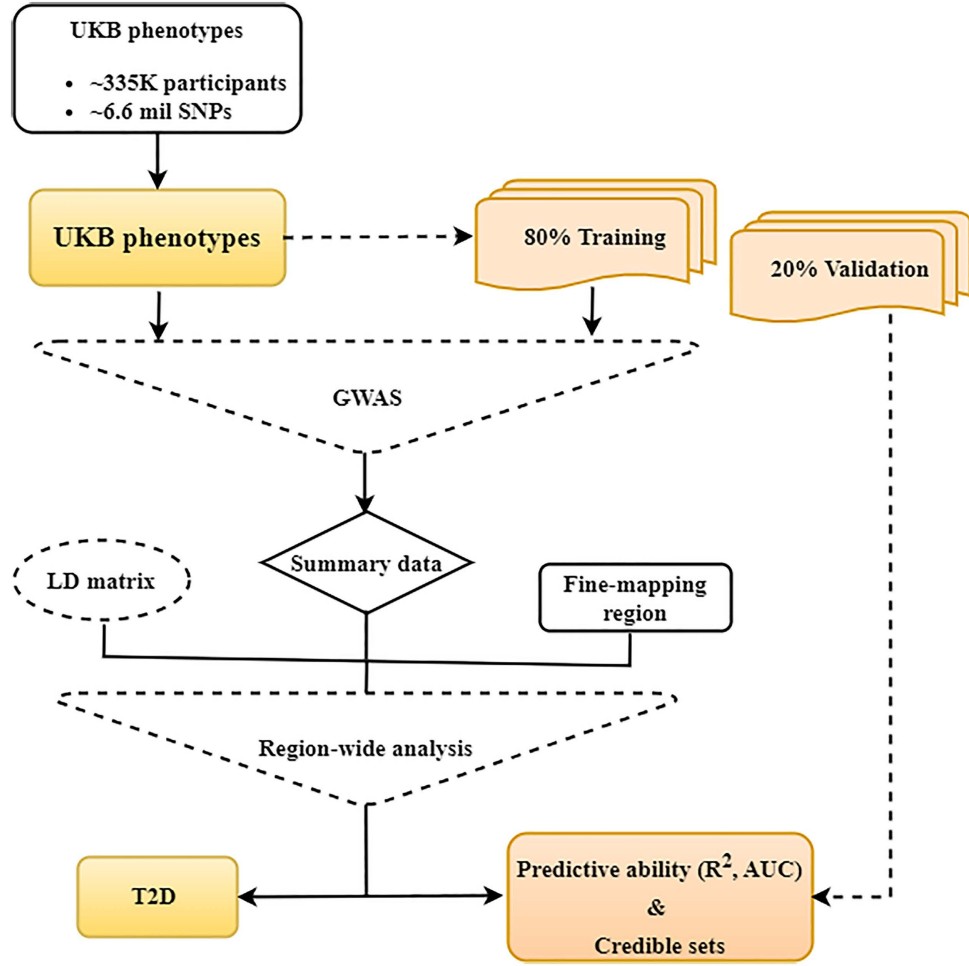

**Fig 2. Flowchart illustrating the design of populations for the analysis of the UK Biobank phenotypes to determine the predictive abilities and features of credible sets across different models.**

priori, LD can induce high posterior correlations. Several priors have been proposed to reflect different genetic architectures. In this study, we focused on the performance of BLR models using two types of priors: BayesC [12] and BayesR [13]. These models jointly estimate marker effects, with the prior influencing how non-causal variants are treated. BayesC uses a spike and slab prior that allows variable selection by assigning a fixed proportion of SNPs ($\pi$) to have zero effect, whereas BayesR extends this by assigning SNPs to multiple effect size categories, enabling both variable selection and varying degrees of shrinkage. Details of the BayesC and BayesR priors, are described below.

**BayesC.** In BayesC [12] the SNP effects, **b**, are a priori assumed to be sampled from a mixture distribution with a point mass at zero and univariate normal distribution conditional on SNP effect variance $\sigma_b^2$. To mimic a scenario where a limited numbers of causal genetic variants influence the trait variation additional variables $\delta_i$ are included to indicate if the $i$-th SNP has an effect or not. Thus, $\delta$ have a prior Bernoulli distribution with the probability $\pi$ of being zero. Therefore, the hierarchy of priors is: (Eqs 7 and 8)

$$p\left(b_i | \delta_i, \sigma_{b_i}^2, \pi\right) = \begin{cases} 0 & \text{with probability } \pi, \\ \sim N\left(0, \sigma_{b_i}^2\right) & \text{with probability } 1 - \pi \end{cases}$$

(7)

$$p\left(\sigma_{b_i}^2|v_b, S_b^2\right) = S_b^2\chi_{v_b}^{-1},\tag{8}$$

where $S_b^2 = \sigma_b^2 v_b$ with $\sigma_b^2 = \frac{\sigma_g^2}{(1-\pi)2\sum_i p_i(1-p_i)}$ because the variance of a $t$ distribution is $\frac{v_b}{v_b-2}$. In the fine-mapping analyses using the BayesC prior, we used initial values of $\pi = (0.01)$.

**BayesR.** In BayesR [13], the SNP effects, **b**, are a priori assumed to be sampled from a mixture distribution with a point mass at zero and univariate normal distributions conditional on common marker effect variance $\sigma_b^2$, and variance scaling factors, $\gamma$ (Eq. 9):

$$b_i|\pi, \sigma_b^2 = \begin{cases} 0 & \text{with probability } \pi_1, \\ \sim N\left(0, \gamma_2\sigma_b^2\right) & \text{with probability } \pi_2, \\ \vdots & \vdots \\ \sim N\left(0, \gamma_C\sigma_b^2\right) & \text{with probability } 1-\sum_{c=1}^{C-1}\pi_c \end{cases}\tag{9}$$

where $\pi = (\pi_1, \pi_2, ...., \pi_C)$ is a vector of prior probabilities and $\gamma = (\gamma_1, \gamma_2, ....., \gamma_C)$ is a vector of variance scaling factors for each of C marker variance classes. The $\gamma$ coefficients are prespecified and constrain how the common SNP effect variance $\sigma_b^2$ scales within each mixture distribution. In the fine-mapping analyses using the BayesR prior, we used initial values of $\gamma = (0, 0.01, 0.1, 1.0)$ and $\pi = (0.99, 0.06, 0.03, 0.01)$.

The prior distribution for the SNP effect variance $\sigma_b^2$ is assumed to be an inverse Chi-square prior distribution, $\chi^{-1}(S_b, \nu_b)$. The proportion of genetic variants in each mixture class ($\pi$) follows a Direchlet $(C, c + \alpha)$ distribution, where $c$ is a vector of length $C$ that contains the counts of the number of variants in each variance class and $\alpha = (1, 1, 1, 1)'$ such that $\pi$ is updated only using information from the data. Using the concept of data augmentation, an indicator variable $d = (d_1, d_2, .., d_{m-1}, d_m)$, is introduced, where $d_j$ indicates whether the effect of the j'th genetic variant is zero or nonzero.

**Genomic region for fine mapping.** For the simulated phenotypes, we designed fine-mapping regions based on the number of SNPs, with a maximum of 1001 SNPs per region. Each region was defined by including approximately 500 SNPs to the left and right of the causal SNP. The number of fine-mapping regions varied depending on the simulation scenario. For simulation scenarios involving two causal variants within a region, we used a larger window, designing regions with up to 2001 SNPs in total. These regions included approximately 1000 SNPs on either side of the causal variants.

For the UKB phenotypes, we define the fine-mapping regions based on the physical position around each lead SNP, i.e., genome-wide significant SNPs ($P$-value $<5 \times 10^{-8}$). We defined a genomic region of one mega base pair (1MB) around the lead SNP. If the regions overlapped by more than 500kb then the regions were merged. This arbitrary number was chosen to limit the size of the regions and assuming that the SNPs added to the region might just increase the size but do not contribute to the analysis.

## Fine mapping using summary statistics

We implemented BayesC and BayesR for fine mapping and compared the performance to the following external methods: FINEMAP [8], SuSiE-RSS [10], SuSiE-Inf and FINEMAP-Inf [6].

**BLR model implementation.** For the simulations, BayesC and BayesR were implemented region-wide and genome-wide, using the R package qgg [21]. This implementation is illustrated in Fig 1. Example scripts for the fine-mapping procedures involving the evaluated BLR models are available at: https://psoerensen.github.io/gact/Document/ Finemapping_bayesian_linear_regression_simulated_data.html.

In the fine-mapping analyses using the BayesC prior we used initial values of $\pi = 0.01$. We evaluated three different implementations: bC1 keeps the prior proportion $\pi$ fixed during model fitting, bC2 updates $\pi$ during the estimation process to better reflect the data. We also included a genome wide version, bCgw, applies the model across the entire genome

rather than within defined fine mapping regions. In the fine-mapping analyses using the BayesR prior, we used an initial value of $\pi = (0.99, 0.06, 0.03, 0.01)$ and evaluated three corresponding implementations: bR1 keeps the prior proportion π fixed during model fitting, bR2 updates π during the estimation process to better reflect the data, bRgw applies the model across the entire genome rather than within defined fine mapping regions.

To apply these models' region-wide, we used GWAS summary statistics and pairwise linkage disequilibrium (LD) information for SNPs within each fine-mapping region. The bC1 and bR1 models treated π as fixed and estimated only the marker effect variance $\sigma_b^2$ and residual variance $\sigma_e^2$. In contrast, bC2 and bR2 treated π as a random variable and estimated it jointly with $\sigma_b^2$ and $\sigma_e^2$ during each iteration.

For genome-wide applications (bCgw and bRgw), we used GWAS summary statistics together with a sparse LD matrix. The sparse LD was estimated using a random sample of 50,000 individuals from the full cohort (n = 335,532, White British ancestry). LD was computed in sliding windows of 2,000 SNPs, shifting one SNP at a time. Only the full estimation model was applied in genome-wide analyses. The PIP-values obtained from these genome-wide BLR models were later used to construct credible sets within the fine-mapping regions.

For the simulation scenarios that contained two causal genetic variants within the fine mapping region, BayesC and BayesR were used region-wide with both model options (bC1/bC2 and bR1/bR2), using GWAS summary statistics and pairwise LD.

**Implementation of external fine-mapping methods.** *SuSiE-RSS model*: The model was applied using the R package susieR [9]. We provided the summary statistics (beta estimates and standard error), the LD information and the sample size. The residual variance was estimated as suggested by the model because in-sample LD was used. We used a default of ten causal SNPs and the default function parameters.

*SuSiE-Inf and FINEMAP-Inf models*: To apply these models, we downloaded python package "run_fine_mapping.py" from the link: https://github.com/FinucaneLab/fine-mapping-inf [6]. We provided the summary statistics (SNP estimates and standard error) along with LD information and the sample size. The number of causal SNPs was assumed to be ten to be consistent with the default number of causal SNPs in susieR. SuSiE-Inf and FINEMAP-Inf models were applied separately. No variance was shared and no priors for the SNPs were provided.

*FINEMAP model*: We downloaded FINEMAP software from the link: http://www.christianbenner.com/finemap_v1.4_x86_64.tgz (v1.4) [8]. We provided the summary statistics (SNP estimates and standard error) along with minor allele frequency (MAF), LD information, and the sample size for the fine mapping regions. The number of causal SNPs was assumed to be ten. No priors for the SNPs were provided.

## Model convergence

To assess the convergence of model parameters, namely $\sigma_b^2, \sigma_g^2, \sigma_e^2$, and π, we used the metric *Z*-score. This involved calculating the difference between the average parameter values taken at the start and end of the iterations. This difference served as our metric to monitor the convergence of the desired parameter. Fine mapping regions with an absolute value of the *Z*-score, for any of the parameters, greater than three was further investigated by thorough evaluation of the trace plots of the parameters. We used the Geweke convergence diagnostic for Markov chains [26]. This diagnostic evaluates whether the means of the first and last segments of the chain (default settings: the first 10% and the last 50%) are equal. If samples are drawn from the stationary distribution, these means should be equal, and the test statistic follows an asymptotically standard normal distribution. The test statistic, a *Z*-score, is calculated as the difference between the two samples means divided by its estimated standard error. We used a threshold of 3.0 (i.e., |*Z*| > 3) to detect non-convergence. This diagnostic was employed for the following model parameters: SNP effect variance ($\sigma_b^2$), residual variance ($\sigma_e^2$) and the proportion of causal variants (π). For densely positioned SNPs, we recommend using multiple independent runs of the Gibbs sampling algorithm. This is done using the nrun parameter in the gmap function from the qgg package (e.g., nrun = 10 will result in an average across ten independent runs).

## Credible sets for simulations

Two different definitions of credible sets (CS) were used in this study, depending on the dataset. CS1 was applied to simulated data, where each fine-mapped region contained only one causal SNP. In this approach, SNPs were sorted by their PIP-value, and those contributing to a cumulative PIP of ≥ 90% formed a credible set [17]. This simplified procedure was used to evaluate methods based on their ability to identify a single causal SNP within a region. CS2, based on prior methods [10,27], was applied to both simulated and real data, potentially involving multiple causal variants per region. In CS2, individual SNPs with PIP ≥ 0.90 were first identified as single-SNP credible sets. Among the remaining SNPs, those in strong linkage disequilibrium (LD) ($r^2 \geq 0.5$) with the SNP having the highest PIP were grouped into credible sets if their cumulative PIP reached ≥ 0.90. This process was repeated iteratively. The goal of CS2 was to assess methods based on their ability to identify multiple causal SNPs within a region. To enable comparison with the original methods, we also applied the credible set procedures implemented in FINEMAP, SuSiE-RSS, and SuSiE-Inf, which allow for multiple credible sets, using parameters similar to those used in CS2 (i.e., coverage of 0.9 and purity of 0.5). A flowchart detailing the CS design process is presented in S1 Fig. Detailed steps utilized to explore the presence of multiple CSs within a fine-mapped region are described in S1 Text.

## Assessment of fine mapping models in simulations

We set out to benchmark the performance of fine-mapping methods using four evaluation metrics: F1 score, recall, precision, and area under the receiver operating characteristic curve (AUC), based on the credible set methods described previously.

The efficiency of different models was evaluated using PIP-values, focusing on AUC and the F1 score, which is the harmonic mean of precision and recall estimated for the credible sets (CS).

To directly compare the models, we quantified their ability to identify causal genetic variants by computing AUC values based on PIPs and an indicator vector denoting whether each variant was truly causal. We report the mean AUC within each simulation scenario across replicates.

In addition, we computed the F1 classification score for the fine-mapping regions based on the credible sets. Each region contained a simulated causal SNP (index SNP). The F1 score ranges from 0 to 1, with values closer to 1 indicating a model's stronger ability to correctly identify true causal SNPs while minimizing false positives:

$$F_1 = \frac{2pr}{p+r},$$

(10)

where, precision, $p = TP/(TP + FP)$ and recall $r = TP/(TP + FN)$.

The F1 score was calculated for each replicate of a simulation scenario. For a given replicate, a true positive (TP) was defined as a credible set (CS) that contained the index SNP for its region (referred to as a true positive CS, or TP). A false positive (FP) was defined as a CS that met the significance threshold (alpha) but did not contain the index SNP (referred to as a false positive CS, or FP). A false negative (FN) was defined as a genomic region where the cumulative sum of PIPs did not exceed the alpha threshold (e.g., 0.9) and no CS was detected.

In addition to this standard definition of FN, we applied two additional criteria. First, regions where a method failed to converge were considered false negatives. Second, TP CSs that contained more than ten SNPs were also considered false negatives, as overly large credible sets provide limited value in pinpointing causal variants.

To assess model efficiency, we examined the number of SNPs in the true positive credible sets (TP), aiming to keep CS size as small as possible. The design of the CSs and the estimation of the F1 score are illustrated in Fig 3.

We also evaluated model performance in fine-mapping regions containing two causal SNPs using the CS2 approach.

**Influence of different factors in simulations.** To investigate the influence of each parameters: $h^2_{SNP}$, π, genetic architectures (*GA*) and *PV* on the performance of the models, we performed TukeyHSD test in R. To quantify the factors

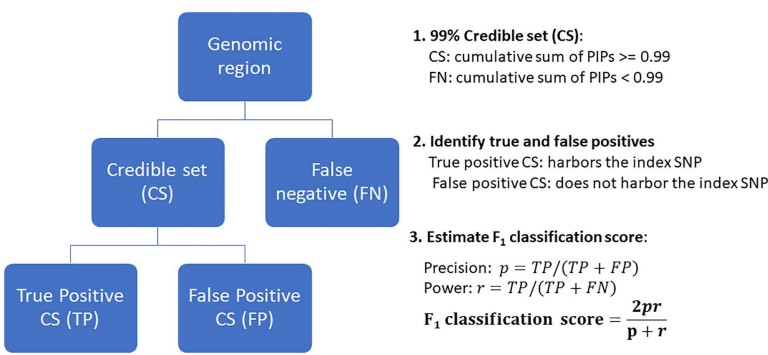

**Fig 3. Design of credible sets with a 0.90 threshold for the cumulative sum of Posterior Inclusion Probabilities (PIPs), and estimation of the F$_1$ classification score based on the credible sets.**

with the greater influence in the simulations, for each model, we also performed ANOVA on the linear model where the performance metric (e.g., $F_1$ score) was regressed on $h^2_{SNP}$, π, and *GA* for the quantitative phenotypes (and *PV* for the binary phenotypes).

## Credible sets for the UKB phenotypes

Unlike simulations where fine-mapping regions were not merged irrespective of overlaps, in the UKB phenotypes, fine-mapping regions were merged if they shared a 500kb overlap of SNPs. This approach increased the likelihood of containing multiple potentially causal SNPs within a single fine-mapped region. To capture multiple potentially causal SNPs, we utilized the CS2 algorithm as described in S1 Text. We applied this CS algorithm across all models in our study and compared the models' performance. For each trait, non-converged fine-mapped regions were excluded across all the models. Afterwards, for each model, we determined the average total number of CSs, the average median CS size (SNP counts in a CS), and the average median value for the average correlations (*avg.r$^2$*) among SNPs in the CS. To estimate *avg.r$^2$*, we excluded the sets with only one SNP as they were not informative, and we used absolute pair-wise correlations among SNPs in the CS. In case the size of CS exceeded 100, only randomly chosen 100 SNPs were used to obtain *avg.r$^2$* for that CS. In a fine-mapped region, SNPs with $PIP_{SNP}$ <= 0.001 was excluded before designing multiple CSs assuming that they would have little to no contribution in meeting the criterion of PIP.

## Assessment of fine mapping models in the UKB phenotypes

**Predictive ability.** For quantitative phenotypes, the predictive ability was determined by estimating the coefficient of determination, ($R^2$). For binary phenotypes, the predictive ability was determined by estimating AUC [28].

Polygenic prediction of observed phenotypic values was performed by calculating a polygenic score (PGS) for each individual in the validation population, for each replicate. The PGS is computed as the sum of the number of effect alleles an individual carries, each weighted by its estimated effect size [29]:

$$PGS = \sum_{i=1}^{m} X_i \hat{b}_i. \tag{11}$$

where $X_i$ refers to the genotype matrix that contains an allelic count and $\hat{b}_i$ is the estimated SNP effect for the *i*-th variant, *m* is the number of SNPs.

To quantify the accuracy of the PGS for real quantitative phenotypes, co-variates adjusted scaled phenotypes for validation population was regressed on the predicted phenotypes. The coefficient of determination, $R^2$, from the regression

was used as a metric to assess the predictive ability of the model. To quantify the accuracy of the PGS for real binary phenotypes, AUC [28] was reported. Difference in the estimates of $R^2$ and AUC (averaged across five replicates) among different methods was compared using TukeyHSD test.

## Application of BLR for fine mapping in T2D

We performed single SNP logistic regression in PLINK 1.9 [23] leveraging the entire UKB cohort for T2D (Table 1), followed by adjustments of the marginal summary statistics with the BayesR model. Fine mapping regions were created as for the UKB phenotypes were defined as described in the S1 Text.

To validate the results obtained from BayesR model for T2D, we conducted non-exhaustive comparison of our findings with the external study. Also, using the R package "gact", we performed a gene set enrichment analysis to identify diseases enriched for T2D-associated genes and tissue-specific expression Quantitative loci (eQTLs) enrichment analysis to identify tissues enriched for T2D.

In the initial step, we mapped SNPs from multiple CSs to genes using the Ensembl Gene Annotation database available at https://ftp.ensembl.org/pub/grch37/release109/gtf/homo_sapiens/Homo_sapiens.GRCh37.87.gtf.gz. This mapping targeted SNPs within the open reading frame (ORF) of a gene, including regions 35kb upstream and 10kb downstream of the ORF, due to their potential regulatory role in controlling main ORF translation.

**Comparison with large-scale meta-GWAS study.** To obtain any overlapping genes in our study with [30], one of the largest and most comprehensive meta-GWAS on T2D. The study consisted of imputed genetic variants from 898,130 European-descent individuals (9% cases). Our study limited comparison to genes given by the study in the S2 Table which provided information of 243 loci (135 newly identified in T2D predisposition) comprising 403 unique genetic signals/associations.

**Gene-diseases association enrichment analysis.** To determine diseases significantly enriched for the gene set of our interest, we first curated a set of genes with PIP of at least 0.5 (sum of *PIP*). We then downloaded the disease-gene associations data from the DISEASE database [31]. This database contained disease–gene association scores (full and filtered) derived from curated knowledge databases, experiments primarily GWAS catalog, and automated text mining of biomedical literature. The analysis was conducted on the final disease-gene association data where association of a gene to a disease was combined from all the above-mentioned sources. This database includes over 10,000 diseases. However, multiple terms in the database were used to refer to the same disease. We investigated enrichment via hypergeometric test [32].

**Tissue-specific eQTLs enrichment analysis.** To determine tissues enriched for eQTLs associated with T2D, firstly multi-tissue cis-eQTL annotation was obtained from GTEx (Genotype-Tissue Expression) consortium (https://storage.googleapis.com/adult-gtex/bulk-qtl/v8/single-tissue-cisqtl/GTEx_Analysis_v8_eQTL.tar) [33]. We identified only eQTLs within our fine-mapped regions for each tissue. We then assessed the enrichment of tissue-specific eQTLs using a multiple linear regression model, adjusting for the influence of other tissue-specific eQTLs. The analysis was conducted using absolute beta-estimates from the BayesR model. The regression model allowed us to calculate Z-scores (coefficient estimates/standard errors) and P-value for each tissue. Tissue-specific eQTLs with a P-value less than 0.05 were considered significantly enriched.

## Results

### Performance in simulation scenarios

Comparison between methods. We evaluated the mean performance (± standard error) of several fine-mapping methods across multiple metrics. Statistical significance was assessed using one-sample *t*-tests comparing each method's mean performance to the overall metric average, as well as ANOVA on a linear model in which the performance

metric (e.g., F1 score) was regressed on method, while adjusting for other simulation design parameters (Figs 4 and S3–S10 Figs). These results were further supported by rank-based analyses (S11 and S12 Figs). To explore the influence of simulation design parameters in more detail, we present F1 scores across all simulation scenarios for both quantitative (S3 Fig) and binary phenotypes (S7 Fig).

Using the simple credible set method (CS1), our results revealed consistent differences in performance across all simulation settings. The bR2 method demonstrated the strongest performance when assessed by F1 classification score (mean = 0.339, $p = 0.004$), significantly outperforming the overall mean (Fig 4A). In contrast, bCgw (mean = 0.261, $p = 0.015$) and FINEMAP-Inf (mean = 0.265, $p = 0.033$) performed significantly worse than average. All other methods, including SuSiE-RSS, SuSiE-Inf, FINEMAP, bC2, and bR1, showed no statistically significant deviations from the mean, suggesting similar F1 performance. In terms of Recall, bR2 again achieved the highest value (0.454, $p < 1 \times 10^{-8}$), followed by bC2 (0.394, $p = 0.013$), both significantly above the mean performance (Fig 4B). bC1 (0.295), bCgw (0.292), and FINEMAP-Inf (0.265) performed significantly below average. The remaining methods showed no significant difference from the overall mean. None of the methods differed significantly in mean precision, indicating comparable ability to identify true positives. Mean values ranged narrowly, from bR2 (lowest at 0.308, $p = 0.264$) to bRgw (highest at 0.334, $p = 0.475$) (Fig 4C). FINEMAP-Inf, SuSiE-RSS, and SuSiE-Inf displayed significantly higher AUC than the average ($p < 0.05$), reflecting stronger classification accuracy (Fig 4D). In contrast, bCgw and bC1 showed significantly lower AUC values, with bCgw performing the worst overall. Other methods, including bR2 and FINEMAP, did not differ significantly from the average.

The rank-based evaluation confirmed that bR2 consistently achieved the highest ranks in both F1 and Recall, with perfect mean values for these metrics, indicating strong and stable performance (S11 Fig). We generally observed a performance trend of bRgw < bR1 < bR2 in both F1 and Recall, except in cases of high polygenicity. Similar pattern was observed among the BayesC-based models. SuSiE-RSS also performed well, particularly in F1 and Precision, while SuSiE-Inf excelled in Precision. FINEMAP showed moderate performance with variability in AUC, whereas FINEMAP-Inf

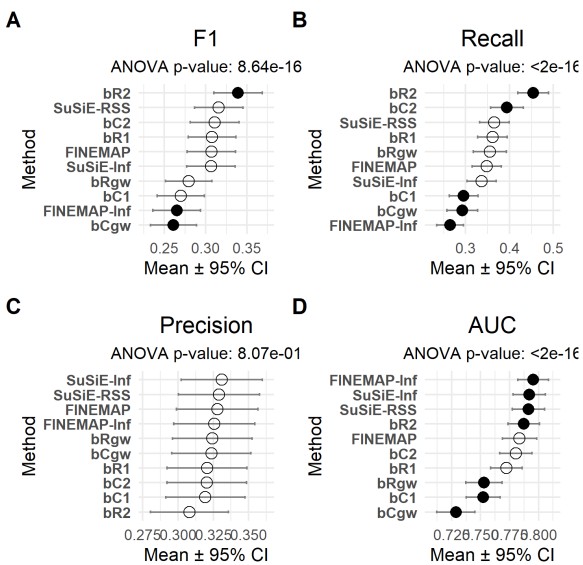

**Fig 4. Method-wise performance (mean ± 95% CI) across four evaluation metrics using the CS1 credible set procedure in one-causal-variant simulation scenarios.** Forest plots display F1-score, Recall, Precision, and AUC for each method. CS1 is applied uniformly to all methods. Points are colored by method and shaped to reflect whether performance differs significantly from the overall mean (p < 0.05). Error bars represent 95% confidence intervals. Methods are ordered by their mean score.

performed poorly in F1 and Recall but ranked well in AUC, suggesting imbalanced performance. bC1 ranked lowest overall, while bC2 and bR1 showed mixed results. Overall, bR2 emerged as the most robust method, followed by SuSiE-RSS and SuSiE-Inf, depending on the metric.

Under the CS1 procedure, credible set sizes were generally small but varied across methods (S2A Fig). SuSiE, SuSiE-Inf, and bR2 had the largest average sizes (2.15–2.23), while bCgw and FINEMAP-Inf produced the smallest (1.34–1.42). Among BLR models, bC1, bCgw, and bRgw yielded the most compact sets. Several methods, including FINEMAP-Inf and multiple BLR models, differed significantly from the overall mean (p < 0.05).

To further enable comparison with the original fine mapping methods, we also applied the credible set procedures implemented in FINEMAP, SuSiE-RSS, and SuSiE-Inf, which allow for multiple credible sets, using parameters similar to those in CS2 (i.e., coverage of 0.9 and purity of 0.5). The overall performance rankings remained largely unchanged. bR2 and bR1 continued to show strong performance, while FINEMAP again underperformed in sparse architectures, although the significance of its underperformance was slightly less consistent (S1 Fig). The differences between SuSiE-RSS and SuSiE-Inf remained minor, while bC1 and bC2 consistently showed lower performance. Under the CS2 procedure, all methods produced small credible sets, with mean sizes ranging from 1.22 (bC1) to 1.79 (FINEMAP) (S2B Fig). SuSiE-based models and BLR methods showed similar performance (means ~1.24–1.29). Compared to CS1, CS2 yielded more compact sets, likely due to LD-based filtering. All methods differed significantly from the overall mean (p < 0.05), indicating consistent differences in set size across methods.

**Impact of simulation parameters.** All models performed best (i.e., achieved the highest F1 scores) under the simulation scenario with moderate SNP heritability ($h^2 = 0.3$), low polygenicity ($\pi = 0.001$), and the GA1 architecture for quantitative phenotypes (S3 Fig), and with a prevalence (PV) of 15 percent for binary phenotypes (S4A Fig).

To evaluate the impact of simulation parameters on fine mapping performance, we conducted an ANOVA with F1 score as the response variable. The analysis revealed that heritability, polygenicity, and genetic architecture all had statistically significant effects on model performance (S3 and S4A Figs). Polygenicity ($\pi$) had the largest influence (F = 27408, $p < 2.2e\text{-}16$); more polygenic scenarios, involving a higher number of small effect variants, resulted in lower F1 scores. This reflects the increased difficulty of accurately identifying many weak signals. Heritability ($h^2$) also had a strong effect (F = 2109, $p < 2.2e\text{-}16$), with higher heritability associated with better performance, indicating that stronger genetic signals improve fine mapping accuracy. Genetic architecture (GA) had a smaller but still significant effect (F = 160, $p < 2.2e\text{-}16$); scenarios with two large effect variants yielded higher F1 scores compared to those with mixed or weak effects.

We also compared performance between continuous and binary trait simulations (S3 and S4A Figs). F1 scores were generally lower for binary traits, indicating the greater challenge of detecting causal variants in dichotomized phenotypes (F = 2489, $p < 2.2e\text{-}16$). Within binary traits, lower prevalence (PV = 0.05) led to poorer performance than higher prevalence (PV = 0.15), likely due to the reduced number of cases and limited statistical power in rare trait scenarios.

**Performance in simulations with two causal variants.** We evaluated the performance of fine-mapping methods using simulations with two causal variants, varying sample sizes (200k, 250k, 300k), and effect size configurations. This analysis used the credible set procedures implemented in FINEMAP, SuSiE-RSS, and SuSiE-Inf, as well as the credible set procedure for the BLR models, which allows for multiple credible sets.

F1 scores improved with increasing sample size across all methods (Fig 5). In the two large-effect scenario (LL), bR1, bR2, bC2, and SuSiE-RSS achieved the highest performance (F1 > 0.75 at n = 300k). In the mixed-effect scenario (LS), bR2 and bC2 maintained relatively strong performance, while all methods struggled under the small-effect (SS) setting. Nonetheless, bR and bC variants generally outperformed SuSiE-Inf and FINEMAP in the SS scenario.

Ranking analyses across all settings showed that bC2 and bR2 achieved the best average F1-score ranks (2.5 and 2.7, respectively), followed by SuSiE-RSS and bR1 (S16 Fig). In contrast, FINEMAP, SuSiE-Inf, and bC1 ranked lowest (mean ranks ~5.0–5.4), indicating weaker overall performance. Most methods performed similarly in terms of Recall (S13 Fig),

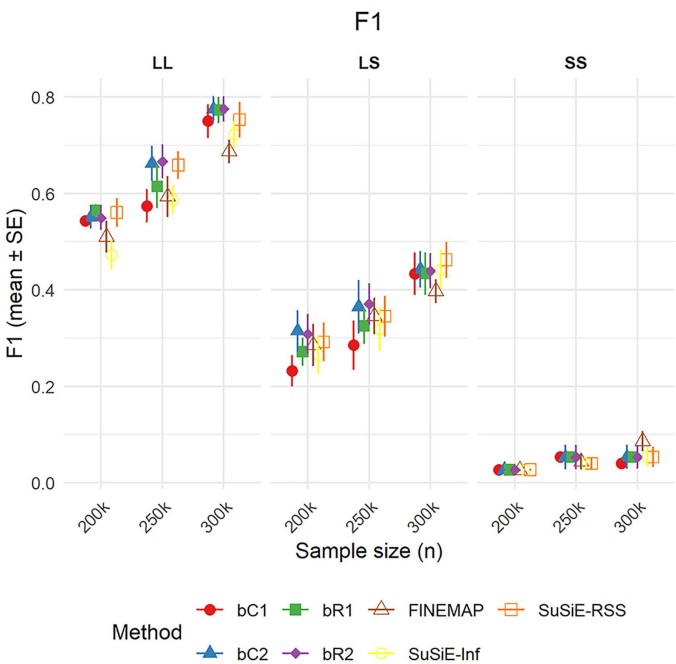

**Fig 5. F1-score performance (mean ± standard error) for fine-mapping methods across sample sizes and causal configurations in two-causal-variant simulation scenarios.** Points represent mean F1-scores by method, with causal configurations grouped as LL (two large effects), LS (one large and one small effect), and SS (two small effects). Error bars show ±1 SE.

Precision (S14 Fig), and AUC (S15 Fig). However, the poorer F1-score performance of FINEMAP is primarily due to its lower Precision (see S6B and S7C Figs).

Across all methods, the average credible set size was similar, ranging from approximately 3.77 to 4.43 variants (S17 Fig). FINEMAP produced the largest average credible sets (mean = 4.43), followed by bR2 (4.20) and bC2 (4.15). SuSiE-Inf yielded the smallest average set size (mean = 3.77), with SuSiE and bR1 also producing relatively small sets (3.82 and 4.02, respectively). These results indicate that, in the presence of two causal variants, all methods produced compact credible sets of comparable size, with minor differences across models.

## Application to UKB phenotypes

**Predictive ability.** We observed a significant decrease in the $R^2_{avg.rep}$ (averaged across all the continuous phenotypes) of BayesC and BayesR relative to SuSiE-Inf and FINEMAP-Inf for the phenotypes BMI, WC, HC and WHR (Fig 6). No significant difference in the $R^2_{avg.rep}$ was observed between BayesR compared to SuSiE-Inf and FINEMAP-Inf for height, whereas a significant decrease was observed for BayesC compared to these models. We observed significant improvement in the $R^2_{avg.rep}$ of BayesR relative to SuSiE-RSS for height. All the methods could predict height better compared to other quantitative phenotypes. Prediction $AUC_{avg.bin}$ (averaged across all the binary phenotypes) with BayesR increased by 0.40%, 0.16%, 0.08%, 0.05% compared to SUSIE-RSS, BayesC, FINEMAP-Inf and SuSiE-Inf, respectively (Fig 7). We did not observe any significant differences between the $AUC_{avg.rep}$ (averaged across all the replicates) of models compared pairwise for any binary phenotypes except for HTN. For HTN, BayesR improved the $AUC_{avg.rep}$ significantly compared to SuSiE-RSS. The highest estimate of the $AUC_{avg.rep}$ was observed for T2D followed by HTN for all the models. The lowest estimate of the $AUC_{avg.rep}$ was observed for RA. Prediction $R^2_{avg.qt}$ (averaged across all the quantitative phenotypes) with BayesR decreased by 5.32% and 3.71% compared to SuSiE-Inf and FINEMAP-Inf, whereas

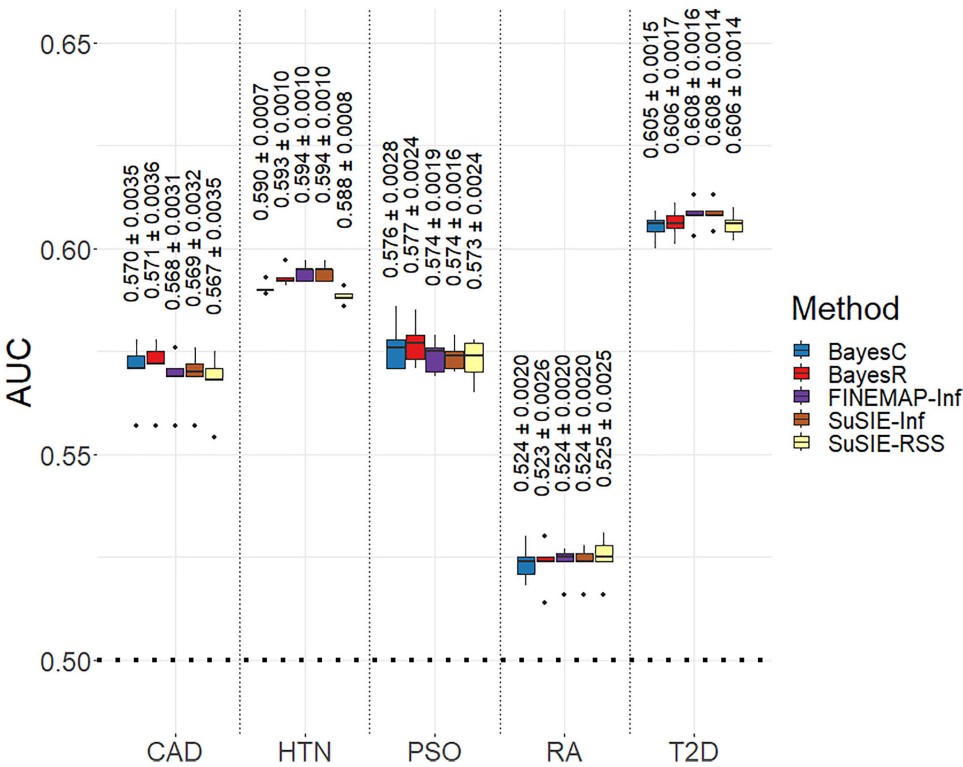

**Fig 6. Prediction accuracies estimated from fine mapped regions.** Box plot of prediction accuracy, represented by the coefficient of determination ($R^2$), averaged across five replicates for the UKB quantitative phenotypes: body mass index (BMI), hip circumference (HC), standing height (Height), waist circumference (WC), and waist-hip ratio (WHR). The models used in the fine mapping can be identified by the colors in the legend associated with each model. For each method within a trait, corresponding mean of R2 or AUC across five replicates and standard error is written on the top of the boxplot.

increased by 7.93% and 8.3% compared to SuSiE-RSS and BayesC. BayesR model improved the $R^2_{avg.rep}$ (averaged across all the replicates) significantly compared to BayesC model for all the quantitative phenotypes except for WHR.

**Credible sets.** The average total number of fine-mapped regions across five replicates for the quantitative phenotypes ranged from 135.2 for WHR to 461 for Height and for the binary phenotypes ranged from 4 for RA to 137.4 for HTN (Table 1). The highest averaged non-converged regions were observed for RA (55%) followed by PSO (29.62%). For other phenotypes, the non-converged regions ranged from 1.22% to 6.42%.

BayesR identified the highest average number of CSs for Height, CAD, HTN and T2D, whereas SuSiE-RSS found the highest average number of CSs for BMI, HC, WC and WHR (Table 1). For the above-mentioned phenotypes, FINEMAP-Inf identified the smallest average number of CSs. All the models obtained a similar average number of CSs for PSO (9–12) and RA (1.8 to 2.2).

The BLR models showed the smallest average median CS size across all the phenotypes compared to the external fine-mapping models (Table 3). BayesR showed the smallest average median size of CS for BMI, Height, WC, WHR, PSO, RA and T2D. BayesC showed the smallest average median size of CS for CAD. Both BayesC and BayesR showed the same average median size for HC and HTN. The highest average median CS size was shown by SuSiE-Inf for BMI, Height, WC, CAD, PSO, RA and T2D. For other phenotypes, SuSiE-RSS showed the highest value for the median CS size.

The average median for *avg.r²* for the BLR models was smaller compared to the external models. BayesC showed the largest average median value compared to BayesR across all the phenotypes.

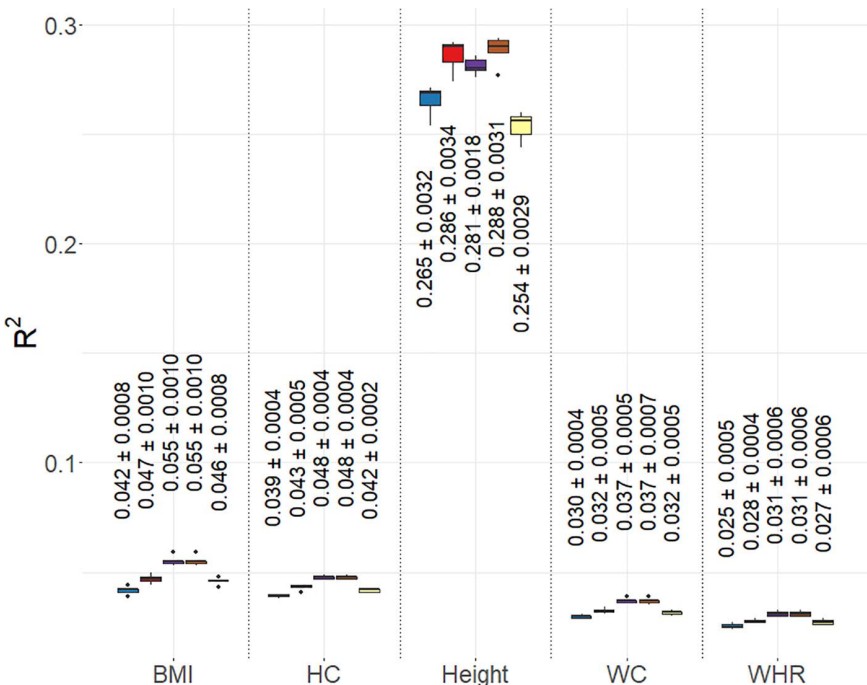

**Fig 7. Prediction accuracies estimated from fine mapped regions.** Box plot of prediction accuracy, represented by the Area under the Curve (AUC), averaged across five replicates for the UKB binary phenotypes: coronary artery disease (CAD), hypertension (HTN), psoriasis (PSO), rheumatoid arthritis (RA), and type 2 diabetes (T2D). The models used in the fine mapping can be identified by the colors in the legend associated with each model. For each method within a trait, corresponding mean of R2 or AUC across five replicates and standard error is written on the top of the boxplot.

## Application of BLR model in T2D

We identified a total of 117 CSs for T2D across 69 fine-mapped regions with a median CS size of 2 (range:1–297), and the median of $avg.r^2$ was 0.80 (range: 0.49 to 1). We identified 53 CSs of size 1 (1 SNP counts), 47 CSs of size between 2–50, and the remaining 17 CSs of size more than 50 SNPs.

**Comparison with large-scale meta-GWAS study.** We found 53 of the 181 genes identified from our study, listed in S2 Table, overlapped with genes from the study Mahajan et al. (2018) [30] (S6 Table). Among 53 overlapped genes, 10 genes (*DTNB, RBM6, MBNL1, SLCO6A1, PDE3B, CELF1, MAP2K7, ZC3H4, EYA2,* and *ZBTB46*) were categorized as novel associations in the study by (26).

Additionally, our study identified multiple SNPs at *TCF7L2* in addition to rs7903146 (PIP: 0.9996). This includes rs34855922 (PIP: 0.3844), rs11196234 (PIP: 0.3512) and rs7912600 (PIP: 0.086) within a CS ($avg.r^2$: 0.70), as well as rs145034729 (PIP: 0.992) linked to *TCF7L2* locus.

**Gene-Diseases association enrichment.** We identified the 30 most significantly enriched diseases ($p < 0.05$) for the T2D-related gene set, highlighting a range of diabetes-related conditions. These include Type 2 Diabetes Mellitus, Diabetes Mellitus, and the ICD10:E11 code used in the UK Biobank. The results also revealed associations with various diabetes subtypes, such as several forms of maturity-onset diabetes of the young (MODY), prediabetes syndrome, gestational diabetes, permanent and transient neonatal diabetes, as well as ICD10-E14 (unspecified T2D) and ICD10-O24 (diabetes in pregnancy). In addition, several other conditions were enriched, including Rheumatoid Arthritis (RA) with ICD10 codes M0, M05, M06, and M069, as well as Wolfram syndrome, hyperglycemia, hyperinsulinism, glucose intolerance, pancreatic agenesis, pancreatic cystadenoma, and insulinoma. Full details are available in S7 Table.

Table 3. Average for the total number of fine mapped regions (FMR) and non-converged regions for the UKB phenotypes along with the total number of credible sets (CSs), median size of the CSs, and median of the average correlations ($r^2$) of the CSs for all the models.

| UKB Phenotypes | Avg. Total FMR | Avg. Non-converged FMR | BayesC | | | BayesR | | | FINEMAP | | | FINEMAP-Inf | | | SUSIE-Inf | | | SUSIE-RSS | | |
|---|---|---|---|---|---|---|---|---|---|---|---|---|---|---|---|---|---|---|---|---|
| | | | Avg. Total CSs | Avg. Med CS size | Avg. Med $r^2$ | Avg. Total CSs | Avg. Med CS size | Avg. Med $r^2$ | Avg. Total CSs | Avg. Med CS size | Avg. Med $r^2$ | Avg. Total CSs | Avg. Med CS size | Avg. Med $r^2$ | Avg. Total CSs | Avg. Med CS size | Avg. Med $r^2$ | Avg. Total CSs | Avg. Med CS size | Avg. Med $r^2$ |
| Body Mass Index (BMI) | 219.8 | 4.8 | 310.6 | 3.4 | 0.93 | 389.8 | 2 | 0.86 | 410.6 | 5 | 0.96 | 89.8 | 8.6 | 0.97 | 204.4 | 20.6 | 0.90 | 447.2 | 15.6 | 0.95 |
| Hip Circumference (HC) | 203 | 5 | 289.4 | 1 | 0.85 | 359.4 | 1 | 0.80 | 348.4 | 2 | 0.98 | 95.6 | 3 | 0.98 | 200.6 | 4.4 | 0.97 | 410.4 | 6.8 | 0.97 |
| Standing Height (Height) | 461 | 29.6 | 1500 | 3.4 | 0.94 | 1846.8 | 2 | 0.87 | 1668 | 7.1 | 0.97 | 513.6 | 8.3 | 0.98 | 721.2 | 17.5 | 0.93 | 1696 | 14.6 | 0.96 |
| Waist Circumference (WC) | 164.4 | 2 | 229.6 | 3.6 | 0.94 | 276.8 | 2 | 0.88 | 299.4 | 6 | 0.97 | 73.2 | 7.7 | 0.98 | 157.4 | 20.8 | 0.92 | 331.6 | 15.6 | 0.96 |
| Waist-Hip Ratio (WHR) | 135.2 | 3.4 | 184.8 | 2.8 | 0.94 | 224.6 | 2 | 0.88 | 238.2 | 5.2 | 0.97 | 77 | 6.6 | 0.98 | 142.6 | 11.4 | 0.93 | 255.8 | 14 | 0.96 |
| Coronary Artery Disease (CAD) | 29.4 | 0.4 | 38.8 | 1.9 | 0.94 | 54.2 | 2.3 | 0.81 | 35.8 | 6.3 | 0.97 | 28 | 5.7 | 0.97 | 35.8 | 8.7 | 0.96 | 44 | 10.3 | 0.97 |
| Hypertension (HTN) | 137.4 | 6.2 | 211 | 2.2 | 0.90 | 277 | 2.2 | 0.80 | 239.8 | 4.2 | 0.97 | 96.8 | 5.2 | 0.98 | 159.4 | 8.8 | 0.96 | 246.8 | 9.8 | 0.96 |
| Psoriasis (PSO) | 10.8 | 3.2 | 10.2 | 3.4 | 0.92 | 9.6 | 1 | 0.89 | 9.8 | 5.3 | 0.98 | 9 | 5.7 | 0.97 | 12 | 10.5 | 0.96 | 11 | 6.4 | 0.98 |
| Rheumatoid Arthritis (RA) | 4 | 2.2 | 1.8 | 1.3 | NA | 1.8 | 1.2 | NA | 5.4 | 1.5 | 1.00 | 1.8 | 2.6 | 0.95 | 2.2 | 3.9 | 0.93 | 2.2 | 2.8 | 0.98 |
| Type 2 Diabetes (T2D) | 49.6 | 1.2 | 62.6 | 1.8 | 0.93 | 81.2 | 1.6 | 0.79 | 70.4 | 4.9 | 0.97 | 47.6 | 6.4 | 0.97 | 60 | 8.5 | 0.96 | 73.2 | 8.1 | 0.97 |

**Tissue-specific eQTLs enrichment.** Among 49 different tissues, significant enrichment ($p < 0.05$) of T2D-related eQTLs were identified in the 13 tissues (S18 Fig): Brain cerebellar hemisphere (n = 419), Cells cultured fibroblasts (n = 718), Brain cerebellum (n = 467), Pituitary (n = 379), Esophagus muscularis (n = 615), Brain nucleus accumbens basal ganglia (n = 308), Lung (n = 624), Skin (not sun exposed suprapubic) (n = 678), Artery tibial (n = 647), Adipose subcutaneous tissue (n = 695), Muscle skeletal tissue (n = 639), Thyroid (n = 810), and Nerve Tibial (n = 804).

## Discussion

Here we aimed to assess the efficiency of BLR models with BayesC and BayesR priors as a fine mapping tool. We applied these models in simulations, and on empirical data from UKB using GWAS summary statistics. BayesC and BayesR models' efficiency were compared to the state-of-the-art methods such as FINEMAP [8], SuSiE-RSS [10], SuSiE-Inf and FINEMAP-Inf [6]. All the models used in our study serve the same purpose of identifying true effects of causal variants. However, they differ in the details in the algorithm and their implementation which applied together can have different impact on the overall performance.

### BLR as a fine mapping methodology

BayesC and BayesR share the same assumptions regarding the prior variance of marker effects but differed in their implementation in our study. Specifically, we compared their performance when applied either genome wide—using all SNPs—or region wide—restricted to predefined fine-mapping regions based on simulated causal variants. To our knowledge, this is the first study to systematically compare BayesC and BayesR in both contexts. We evaluated several variants of BLR models, which differed along two main dimensions: (1) whether the prior proportion of non-zero effects ($\pi$) was fixed or estimated, and (2) whether the model was applied at the region level or genome wide. For BayesR, bR1 estimates $\pi$ from the data, while bR2 uses a fixed value during model fitting. Similarly, for BayesC, bC1 estimates $\pi$, whereas bC2 uses a fixed $\pi$. Overall, bC2 and bR2 outperformed bC1 and bR1 in terms of F1 classification score and Recall, suggesting that using a fixed $\pi$ may improve performance under certain conditions. All models estimated the marker effect variance. A smaller estimate of this variance can lead to better model fit either by assigning small effects to a larger number of SNPs or by concentrating larger effects on fewer SNPs, depending on the underlying genetic architecture. Further investigation is needed to determine whether improved control of hyperparameters, particularly those governing $\pi$ and marker effect variance, can enhance fine mapping accuracy.

To further explore the effect of modeling scope, we also included genome-wide versions of the models: bCgw and bRgw, which analyze the entire genome in a single model rather than being restricted to predefined fine-mapping regions. We generally observed a performance trend of bRgw < bR1 < bR2 in both F1 and Recall, except in cases of high polygenicity; a similar pattern was observed among the BayesC-based models. This improvement in Recall and F1 appeared to come at a slight cost to Precision, although the differences were not statistically significant compared to the overall mean.

Together, this suggests that applying BLR models at the region level can improve fine mapping resolution by providing better estimates of local marker effect variances. In contrast, genome-wide models may be more appropriate for highly polygenic traits or for settings where regional boundaries are poorly defined, and they tended to improve Precision. Our framework allowed us to systematically evaluate how the choice of prior specification, the treatment of $\pi$, and the modeling scope influence the accuracy and efficiency of fine mapping.

### Comparison of the BLR models to external models

Our results demonstrate that Bayesian Linear Regression models, particularly bR2, can outperform widely used fine-mapping methods such as FINEMAP and SuSiE-RSS under a range of simulation scenarios. bR2 consistently showed superior performance in F1 score and Recall, indicating a better balance between sensitivity and precision when identifying true causal variants. In contrast, external methods such as FINEMAP-Inf and SuSiE-Inf often showed imbalanced

performance, strong in AUC but weaker in Recall, suggesting limitations in identifying true positives despite good overall classification.

Among the BLR variants, models with fixed π (bR2, bC2) generally outperformed those that estimate π from the data (bR1, bC1), suggesting that fixed priors may improve model stability and fine-mapping accuracy. In particular, bC1 consistently ranked lowest among all methods, while bC2 showed moderate but improved performance. We used in-sample LD, while using external summary statistics in-sample LD is not always available [34]. Hence, the recall may decrease while using an external reference LD panel.

While SuSiE-RSS and SuSiE-Inf performed competitively in Precision and AUC, they did not consistently outperform the best BLR models. Similarly, FINEMAP showed moderate performance overall but underperformed in sparse genetic architectures.

Our findings were robust to different credible set procedures, with performance rankings remaining consistent across both simple (CS1) and multiple-set (CS2-like) methods. While bR2 maintained strong performance, bR1 and bC2 also demonstrated reliability across metrics, suggesting they may serve as viable alternatives depending on the analysis context.

When estimating Recall (and F1 classification scores), in addition to requiring that a set surpasses a cumulative PIP threshold of 0.90 to be considered a credible set, any credible set containing a causal SNP but exceeding 10 SNPs in size was treated as a false negative (FN), thereby penalizing overly large sets. While this threshold is strict and somewhat arbitrary, it reflects the limited value of large CSs in sparse genetic data. Results may vary with different thresholds, and it would be interesting to assess model performance under alternative CS size cutoffs.

In simulations with two causal variants, BLR models, particularly bC2 and bR2, consistently demonstrated strong and reliable performance across varying sample sizes and effect-size configurations. While all methods benefited from larger sample sizes, BLR models maintained an advantage even in challenging scenarios with small-effect variants, where external methods like FINEMAP and SuSiE-Inf underperformed. These results highlight the robustness of BLR models with fixed π, especially in sparse or heterogeneous genetic architectures, and support their use in diverse fine-mapping contexts.

These findings highlight the advantages of BLR models, especially bR2, in scenarios where accurate identification of causal variants is critical. They also emphasize the importance of modeling assumptions, such as the treatment of π and model scope, in influencing fine-mapping outcomes. Our results suggest that BLR models can serve as strong alternatives, or even preferred options, to established external methods in fine mapping applications.

To our knowledge, this is the first study comparing the BayesR model to leading fine mapping methods, including FINEMAP, SuSiE-RSS, SuSiE-Inf, and FINEMAP-Inf. A previous study by de Los Campos et al. [34] evaluated BayesC against various methods, including SuSiE-RSS and FINEMAP, and found that BayesC performed comparably to SuSiE-RSS and outperformed FINEMAP in terms of Recall and false discovery rate across multiple simulation scenarios. Our results are consistent with those findings. There are, however, differences in study design. The previous study applied models at the whole-genome level using a local regression approach, whereas our analysis was restricted to specific regions defined by simulated causal SNPs, which did not necessarily span the entire genome. Additionally, we compared SuSiE-RSS (an extension of SuSiE that operates on GWAS summary statistics) along with BayesC and FINEMAP, using summary statistics with in-sample LD. In contrast, de Los Campos et al. [34] used individual-level data for BayesC and SuSiE, and summary statistics with in-sample LD only for FINEMAP. Despite these differences in data and scope, we did not expect, nor observe, major discrepancies in results.

## Credible set size comparison

Across both CS1 and CS2 procedures, BLR models (bC1, bC2, bCgw, bR1, bR2, bRgw) produced comparable or more compact credible sets than external methods (FINEMAP, FINEMAP-Inf, SuSiE, SuSiE-Inf, and SuSiE-RSS) (S2 and S17 Figs).

In regions with one causal variant, CS1, based solely on PIP ranking, resulted in the smallest credible sets for BLR models such as bC1, bCgw, and bRgw (mean size 1.34–1.52), while external methods like SuSiE and SuSiE-Inf produced larger sets (2.15–2.23). Under CS2, which supports multiple credible sets per region and applies LD-based filtering (e.g., purity ≥ 0.5), all methods produced smaller sets overall (1.22–1.79). However, the size advantage of BLR models persisted, with bC1 and bR2 producing some of the most compact sets (1.22 and 1.29, respectively), while FINEMAP yielded the largest (1.79).

In regions with two causal variants, credible set sizes increased slightly across methods (3.77–4.43), but similar patterns remained. bR1 and SuSiE-based methods produced the smallest sets, while FINEMAP, bR2, and bC2 generated the largest. These results demonstrate that BLR models consistently produce concise credible sets, particularly when using LD-aware procedures like CS2, while maintaining competitive fine-mapping performance.

### Influence of parameters in simulations

Simulation design parameters had a strong influence on fine mapping performance. Across models, the best results were observed under moderate heritability ($h^2 = 0.3$), low polygenicity ($\pi = 0.001$), and simple genetic architectures (GA$_1$), particularly for continuous traits and binary traits with higher prevalence (PV = 15%). In contrast, scenarios with low heritability, high polygenicity, or complex genetic architectures led to significantly lower F1 scores.

ANOVA confirmed that polygenicity had the largest impact on performance, followed by heritability and genetic architecture. The effect of polygenicity likely reflects the increased difficulty of detecting many small-effect variants, which contribute weak signals and reduce the precision of credible sets. As expected, scenarios with fewer causal variants sampled from a larger marker effect variance produced stronger signals and higher PIPs, making causal SNPs more likely to be captured in credible sets. These findings underscore the importance of genetic architecture and trait characteristics—such as heritability, prevalence, and polygenicity, when evaluating or applying fine mapping methods. They also highlight the need to consider simulation design carefully when benchmarking models or interpreting results.

### The UKB phenotypes, accuracy and fine mapping, credible sets

BayesR showed significantly higher prediction accuracy ($R^2$) than BayesC for four out of five quantitative phenotypes. These findings are consistent with previous studies. Zhu *et al.* reported improved prediction ability using BayesR over BayesC across several economically important cattle traits [35], while Mollandin *et al.* observed similar results in simulations for traits with high heritability [36]. In our analysis, BayesR also produced more, and smaller credible sets compared to BayesC, suggesting that its assumptions about genetic architecture may be better suited for predicting polygenic traits where a range of effect sizes is present.

Despite this, the average prediction accuracy for both BayesR and BayesC was significantly lower than that of SuSiE-Inf and FINEMAP-Inf for traits like BMI, HC, WC, and WHR. The higher prediction performance of the infinitesimal models may be explained by our use of SNP effects from both the sparse and infinitesimal components in SuSiE-Inf and FINEMAP-Inf. This contrasts with a previous study [6], where only the sparse components were used to compare prediction accuracy. Because infinitesimal components account for small effects across all SNPs, they may better capture the genetic signal underlying complex traits. Notably, BayesR's performance approached that of the infinitesimal models, reinforcing its suitability for modeling polygenic architectures.

The predictive accuracies we observed for UK Biobank phenotypes were lower than those reported in other studies. For traits such as BMI, height, HC, WHR, and T2D, the $R^2$ values reported by Lloyd-Jones *et al.* using SBayesR and approximately 1.1 million SNPs were notably higher than those obtained with BayesR in our analysis [14]. This discrepancy is likely due to differences in SNP coverage and model scope. Our predictions were based only on imputed SNPs within fine mapped regions, whereas previous studies used genome-wide SNP sets.

Additionally, for highly polygenic traits like height and BMI, prior work [37] has shown enrichment of heritability from rare variants (MAF < 0.01), which we excluded in our study to focus on common SNPs. Moreover, we excluded non-converged regions from our prediction and credible set analyses, which may have further impacted the overall predictive accuracy.

## Validation of BLR model

Compared to the recent meta-GWAS on T2D [30], we identified 10 genes to overlap with the 53 genes, which were categorized as novel loci in [30]. This finding demonstrates the effectiveness of BayesR model combined with credible sets in identifying potential causal variants, even in studies with comparatively smaller size. This limited number of overlapping genes could be attributed to our study's smaller scale (25,828 cases and 309,704 controls compared to 74,124 cases and 824,006 controls in [30]), which could limit ability to detect especially rare variants, and the exclusion of rare variants (excluding SNPs with < 1% MAF in our study). Additionally, the discrepancies in how SNPs were mapped to a gene between our study and that of [30] might also contribute to this limited overlap.

*TCF7L2* (Transcription Factor 7-like 2) explained the highest genetic variance (0.035) in our study. This gene plays a crucial role in Wnt signaling pathway, which regulates pancreatic islet cell proliferation and survival [38]. In *TCF7L2*, rs7903146 is the largest-effect common variant signal for T2D in Europeans [30]. Observation of multiple signals for T2D at *TCF7L2* in addition to rs7903146 in [30] was the first evidence according to this study. In addition to the rs7903146, we also identified SNP rs34855922 associated to T2D similar with [30], which again demonstrates the effectiveness of BayesR model combined with CSs. The rs7903146 and rs34855922 are two of the eight SNPs that mark regulatory elements within *TCF7L2* locus [39]. The rs7903146 coordinate regulation of *TCF7L2* expression and overlaps histone modification marks and an annotated enhancer in the pancreas [39]. Our study also identified an intronic variant (rs145034729) at the *TCF7L2* locus. The effect of this intronic SNP is uncertain. However, it may function as an enhancer element, modulating the expression of distal genes without necessarily affecting the function of *TCF7L2* itself. The discovery of multiple variants within the *TCF7L2* locus is interesting, as Nyaga et al., suggests that it acts as a regulatory hub for genes implicated in the etiology of T2D [39]. Identifying these variants in this locus offers valuable insights into the biological mechanisms underlying T2D.

The gene set enrichment analysis for diseases provided further support for the efficacy of BayesR model in T2D. This analysis revealed significant enrichment of our gene set for diseases such as T2D, hyperglycemia (diabetes-like symptoms), hyperinsulinism (one of the processes leading to hyperglycemia [40]. Significant enrichment to other types of diabetes and diseases may reflect shared genetic factors (via pleiotropic genes or common pathways) influencing the etiology of diverse conditions (diseases) through different mechanisms. For instance, Tian et al., noted an increased risk of diabetes mellitus incidence in individuals with RA [41], highlighting the potential role of inflammatory pathways in the T2D pathogenesis.

For tissue enrichment analysis, our findings indicate that T2D related eQTLs exhibit tissue-specific effects on gene expression. The implications of our results can be viewed from multiple perspectives. Our results may suggest a complex interplay of regulatory regions in significantly enriched tissues leading to T2D predisposition. Our results may also suggest individuals with T2D might experience adverse effects in these tissues, potentially leading to a range of complications. For instance, Hemerich *et al.*, explored the association of significantly enriched tissue specific T2D associated eQTLs with different T2D complications [42]. Here we delve into the cerebellar hemisphere region of the brain, the most significant enriched tissue. This region, part of the cerebellum (another significant tissue in our study), has been linked to cognitive impairments when abnormal. Roy *et al.*, highlighted significant cognitive impairments in T2D individuals [43], correlating these deficits with considerable loss in gray matter volume in brain regions associated with these functions. The decline in insulin transport and resistance in the cerebral cortex, an area dense with high insulin receptor, may impair regional glucose metabolisms, leading to gray matter volume changes potentially leading to structural and functional changes in brain in T2D individuals.

No association with pancreatic tissue was found, likely due to the GTEx database's limitations. The pancreatic tissue in GTEx represents mostly (97%) exocrine cells that mask islets signals [44]. Pancreatic islets are clusters of specialized endocrine cells that are essential to maintain glucose homeostasis and play a central role in etiology of T2D.

Our study was confined to the cis-eQTLs database from GTEx consortium. Torres et al., have shown that trans-eQTLs contribute significantly to T2D heritability, suggesting that further exploration of trans-eQTLs could enhance the understanding of gene expression and cellular functions across different tissues [45].

In conclusion, we observed that the performance of the BLR models was comparable to the state-of-the-art external models. The performance of BayesR prior was closely aligned with SuSiE-Inf and FINEMAP-Inf models. Results from both simulations and application of the models in the UKB phenotypes suggest that the BLR models are efficient fine mapping tools.

## Supporting information

**S1 Text. Design of multiple credible sets in a fine-mapping region for the UKB phenotypes.**
(DOCX)

**S1 Fig. Method-wise performance (mean±95% CI) across four evaluation metrics using the CS2 credible set procedure in one-causal fine-mapping regions.** Forest plots display the mean performance (±95% confidence intervals) of each fine-mapping method across four metrics: F1-score (panel A), Recall (panel B), Precision (panel C), and AUC (panel D). The CS2 credible set procedure is used for the BLR models, while FINEMAP, SuSiE-RSS, and SuSiE-Inf use their respective internal procedures. Each panel shows the mean and 95% confidence interval of the specified metric across simulation replicates. Points are color-coded to indicate whether a method's performance significantly differs ($p < 0.05$, based on a one-sample t-test) from the overall mean across all methods. Solid black points indicate statistically significant deviations; hollow circles indicate non-significant differences. Horizontal error bars represent the 95% confidence intervals. Methods are sorted by their mean score, with higher values indicating better performance.
(TIF)

**S2 Fig. Credible set size (mean±95% CI) across fine-mapping methods using CS1 and CS2 procedures in one-causal fine-mapping regions.** Forest plots show method-wise mean credible set size (±95% confidence intervals) across simulation replicates. Panel A shows results from one-causal regions using the CS1 credible set procedure applied to all methods. Panel B also reflects one-causal regions but uses the CS2 procedure for BLR models and the internal credible set procedures of FINEMAP, SuSiE-RSS, and SuSiE-Inf. Points are color-coded to indicate whether a method's performance significantly differs ($p < 0.05$, based on a one-sample t-test) from the overall mean across methods. Solid black points indicate statistically significant deviations; hollow circles indicate non-significant differences. Horizontal error bars represent the 95% confidence intervals. Methods are sorted by their mean credible set size, with lower values indicating better performance.
(TIF)

**S3 Fig. F1 classification score performance (mean±standard error) of fine-mapping methods for continous traits across simulation settings in one-causal fine-mapping regions.** Results are stratified by heritability ($h^2$), proportion of causal variants ($\pi$), and genetic architecture (GA). Each panel represents a distinct value of $\pi$, with rows corresponding to genetic architecture (GA1 and GA2). Data points indicate the mean AUC performance (±SE) of each method across simulation replicates. Solid shapes denote the bC and bR methods; hollow shapes represent SuSiE and FINEMAP variants. Higher values indicate better performance.
(TIF)

**S4 Fig. Recall performance (mean±standard error) of fine-mapping methods for continous traits across simulation settings in one-causal fine-mapping regions.** Results are stratified by heritability ($h^2$), proportion of causal

variants ($\pi$), and genetic architecture (GA). Each panel represents a distinct value of $\pi$, with rows corresponding to genetic architecture (GA1 and GA2). Data points indicate the mean AUC performance (±SE) of each method across simulation replicates. Solid shapes denote the bC and bR methods; hollow shapes represent SuSiE and FINEMAP variants. Higher values indicate better performance.
(TIF)

**S5 Fig. Precision performance (mean ± standard error) of fine-mapping methods for continous traits across simulation settings in one-causal fine-mapping regions.** Results are stratified by heritability ($h^2$), proportion of causal variants ($\pi$), and genetic architecture (GA). Each panel represents a distinct value of $\pi$, with rows corresponding to genetic architecture (GA1 and GA2). Data points indicate the mean AUC performance (±SE) of each method across simulation replicates. Solid shapes denote the bC and bR methods; hollow shapes represent SuSiE and FINEMAP variants. Higher values indicate better performance.
(TIF)

**S6 Fig. AUC performance (mean ± standard error) of fine-mapping methods for continous traits across simulation settings in one-causal fine-mapping regions.** Results are stratified by heritability ($h^2$), proportion of causal variants ($\pi$), genetic architecture (GA), and disease prevalence ($p_v$). Each panel represents a distinct value of $\pi$, with rows corresponding to genetic architecture (GA1 and GA2). Data points indicate the mean AUC performance (±SE) of each method across simulation replicates. Solid shapes denote the bC and bR methods; hollow shapes represent SuSiE and FINEMAP variants. Higher values indicate better performance.
(TIF)

**S7 Fig. F1 classification score performance (mean ± standard error) of fine-mapping methods for binary traits across simulation settings in one-causal fine-mapping regions.** Results are stratified by heritability ($h^2$), proportion of causal variants ($\pi$), genetic architecture (GA), and disease prevalence ($p_v$). Each panel represents a distinct value of $\pi$, with rows corresponding to combinations of prevalence ($p_v$ = 0.05 or 0.15) and genetic architecture (GA1 or GA2). Data points indicate the mean AUC performance (±SE) of each method across simulation replicates. Solid shapes denote the bC and bR methods; hollow shapes represent SuSiE and FINEMAP variants. Higher values indicate better performance.
(TIF)

**S8 Fig. Recall performance (mean ± standard error) of fine-mapping methods for binary traits across simulation settings in one-causal fine-mapping regions.** Results are stratified by heritability ($h^2$), proportion of causal variants ($\pi$), genetic architecture (GA), and disease prevalence ($p_v$). Each panel represents a distinct value of $\pi$, with rows corresponding to combinations of prevalence ($p_v$ = 0.05 or 0.15) and genetic architecture (GA1 or GA2). Data points indicate the mean AUC performance (±SE) of each method across simulation replicates. Solid shapes denote the bC and bR methods; hollow shapes represent SuSiE and FINEMAP variants. Higher values indicate better performance.
(TIF)

**S9 Fig. Precision performance (mean ± standard error) of fine-mapping methods for binary traits across simulation settings in one-causal fine-mapping regions.** Results are stratified by heritability ($h^2$), proportion of causal variants ($\pi$), genetic architecture (GA), and disease prevalence ($p_v$). Each panel represents a distinct value of $\pi$, with rows corresponding to combinations of prevalence ($p_v$ = 0.05 or 0.15) and genetic architecture (GA1 or GA2). Data points indicate the mean AUC performance (±SE) of each method across simulation replicates. Solid shapes denote the bC and bR methods; hollow shapes represent SuSiE and FINEMAP variants. Higher values indicate better performance.
(TIF)

**S10 Fig. AUC performance (mean±standard error) of fine-mapping methods for binary traits across simulation settings in one-causal fine-mapping regions.** Results are stratified by heritability ($h^2$), proportion of causal variants ($\pi$), genetic architecture (GA), and disease prevalence ($p_v$). Each panel represents a distinct value of $\pi$, with rows corresponding to combinations of prevalence ($p_v$ = 0.05 or 0.15) and genetic architecture (GA1 or GA2). Data points indicate the mean AUC performance (±SE) of each method across simulation replicates. Solid shapes denote the bC and bR methods; hollow shapes represent SuSiE and FINEMAP variants. Higher values indicate better performance. (TIF)

**S11 Fig. Average performance ranks of fine-mapping methods across simulation settings using the CS1 credible set procedure in one-causal fine-mapping regions.** Mean rank±standard error (SE) of method performance across simulation settings for four evaluation metrics using credible set method CS1 (simple approach). This multi-panel figure shows the mean rank±standard error (SE) for each fine-mapping method based on four metrics: F1-score (panel A), Recall (panel B), Precision (panel C), and AUC (panel D). For each metric, methods were ranked within each simulation setting defined by combinations of genetic architecture (GA), polygenicity ($\pi$), and heritability ($h^2$). Lower ranks indicate better performance. The ranks were then averaged across all settings, and the standard error was computed to reflect variability in ranks. Each point in the plot represents the average rank of a method, with error bars indicating ±1 SE. Methods that consistently rank higher across simulations appear lower on the vertical axis. This visualization allows a direct comparison of method stability and performance across multiple evaluation criteria. (TIF)

**S12 Fig. Average performance ranks of fine-mapping methods across simulation settings using the multiple credible set procedures in one-causal fine-mapping regions.** Mean rank±standard error (SE) of method performance across simulation settings for four evaluation metrics using credible set method CS2 (advanced approach).This multi-panel figure shows the mean rank±standard error (SE) for each fine-mapping method based on four metrics: F1-score (panel A), Recall (panel B), Precision (panel C), and AUC (panel D). For each metric, methods were ranked within each simulation setting defined by combinations of genetic architecture (GA), polygenicity ($\pi$), and heritability ($h^2$). Lower ranks indicate better performance. The ranks were then averaged across all settings, and the standard error was computed to reflect variability in ranks. Each point in the plot represents the average rank of a method, with error bars indicating ±1 SE. Methods that consistently rank higher across simulations appear lower on the vertical axis. This visualization allows a direct comparison of method stability and performance across multiple evaluation criteria. (TIF)

**S13 Fig. Recall performance (mean±standard error) across fine-mapping methods, sample sizes, and genetic architectures in two-causal fine-mapping regions.** Each point represents the mean Recall of a method (indicated by color and shape) across simulation replicates for a given sample size (n = 200k, 250k, 300k) and causal configuration. Causal configurations are grouped as LL (two large-effect causal variants), LS (one large and one small effect), and SS (two small-effect variants). Error bars indicate ±1 standard error. Higher values indicate better performance. (TIF)

**S14 Fig. Precision performance (mean±standard error) across fine-mapping methods, sample sizes, and genetic architectures in two-causal fine-mapping regions.** Each point represents the mean Recall of a method (indicated by color and shape) across simulation replicates for a given sample size (n = 200k, 250k, 300k) and causal configuration. Causal configurations are grouped as LL (two large-effect causal variants), LS (one large and one small effect), and SS (two small-effect variants). Error bars indicate ±1 standard error. Higher values indicate better performance. (TIF)

**S15 Fig. AUC performance (mean±standard error) across fine-mapping methods, sample sizes, and genetic architectures in two-causal fine-mapping regions.** Each point represents the mean Recall of a method (indicated by

color and shape) across simulation replicates for a given sample size (n = 200k, 250k, 300k) and causal configuration. Causal configurations are grouped as LL (two large-effect causal variants), LS (one large and one small effect), and SS (two small-effect variants). Error bars indicate ±1 standard error. Higher values indicate better performance.
(TIF)

**S16 Fig. Comparative ranking of fine-mapping methods across simulation settings and evaluation metrics in two-causal fine-mapping regions.** This multi-panel figure presents the mean performance rank (± standard error) of each fine-mapping method across four evaluation metrics: F1-score (panel A), Recall (panel B), Precision (panel C), and AUC (panel D). For each metric, methods were ranked within each combination of sample size and causal configuration, with lower ranks indicating better performance. The figure shows the average rank and corresponding standard error across all sample sizes, with panels faceted by causal architecture. Causal configurations are grouped as LL (two large-effect causal variants), LS (one large- and one small-effect variant), and SS (two small-effect variants). Error bars indicate ±1 standard error. Lower mean ranks reflect better overall performance.
(TIF)

**S17 Fig. Credible set size (mean ± 95% CI) across fine-mapping methods using CS2 procedures in two-causal fine-mapping regions.** Forest plots show method-wise mean credible set size (±95% confidence intervals) across simulation replicates. The CS2 credible set procedure for the BLR models and the internal credible set procedures implemented in FINEMAP, SuSiE-RSS, and SuSiE-Inf. Points are color-coded to indicate whether a method's performance significantly differs ($p < 0.05$, based on a one-sample t-test) from the overall mean across all methods. Solid black points indicate statistically significant deviations; hollow circles indicate non-significant differences. Horizontal error bars represent the 95% confidence intervals. Methods are sorted by their mean score, with lower values indicating better performance.
(TIF)

**S18 Fig. Significance values for enrichment analysis of tissue-specific eQTLs.** The black dashed line represents the significance cut-off (p-value < 0.05). The size of the points corresponds to the number of eQTLs (eQTL counts) in the tissue.
(PNG)

**S1 Table. Values for the parameters such as heritability, proportion of causal genetic variants, genetic architecture, and prevalence leading to different simulation scenarios for the quantitative and the binary phenotypes.** S2–S5 Tables contain the processed fine-mapping results for all methods across all simulations and can be used to reproduce the figures and results presented in this revised manuscript.
(XLSX)

**S2 Table. Performance metrics for fine-mapping methods across simulation replicates.** For each replicate, the number of true positives (TP), false positives (FP), and false negatives (FN) is shown, along with derived metrics: Precision, Recall, F1-score, and AUC. Simulations were stratified by genetic architecture (GA), heritability ($h^2$), and polygenicity ($\pi$).
(XLSX)

**S3 Table. Credible set counts and average sizes across simulation replicates for all fine-mapping methods using credible set procedure CS1.**
(XLSX)

**S4 Table. Credible set counts and average sizes across simulation replicates for all fine-mapping methods, using the CS2 procedure for BLR models and internal credible set definitions for external methods.**
(XLSX)

**S5 Table. Performance metrics and credible set characteristics for fine-mapping methods across simulation replicates with two causal variants.** For each replicate, the number of true positives (TP), false positives (FP), and false negatives (FN) is shown, along with derived metrics: Precision, Recall, F1-score, and AUC. Credible set counts and average sizes are also reported, using the CS2 procedure for BLR models and the internal credible set definitions for external methods.
(XLSX)

**S6 Table. List of genes from Type 2 diabetes' credible sets, along with their position (chromosome, start and stop position), posterior inclusion probability (PIP), genetic variance (GenVar), and Overlap.** The Overlap column indicates if a gene is also located by the study Mahajan et al. 2018 [30].
(XLSX)

**S7 Table. Top 30 diseases, from DISEASE database, associated with Type 2 diabetes' gene sets.**
(XLSX)

## Acknowledgments

This research has been conducted using the UK Biobank Resource under application number 96479.

## Author contributions

**Conceptualization:** Merina Shrestha, Palle Duun Rohde, Peter Sørensen.

**Data curation:** Merina Shrestha, Zhonghao Bai.

**Formal analysis:** Merina Shrestha, Zhonghao Bai, Peter Sørensen.

**Funding acquisition:** Mads Kjolby, Palle Duun Rohde, Peter Sørensen.

**Methodology:** Merina Shrestha, Palle Duun Rohde, Peter Sørensen.

**Project administration:** Peter Sørensen.

**Software:** Peter Sørensen.

**Supervision:** Palle Duun Rohde, Peter Sørensen.

**Visualization:** Merina Shrestha, Peter Sørensen.

**Writing – original draft:** Merina Shrestha, Palle Duun Rohde, Peter Sørensen.

**Writing – review & editing:** Merina Shrestha, Zhonghao Bai, Tahereh Gholipourshahraki, Astrid J. Hjelholt, Sile Hu, Mads Kjolby, Palle Duun Rohde, Peter Sørensen.

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
