## [Decision Letter · Decision Letter 0]

Dear Dr Shrestha,

Thank you very much for submitting your Research Article entitled 'Evaluation of Bayesian Linear Regression Models as a Fine Mapping tool' to PLOS Genetics.

The manuscript was fully evaluated at the editorial level and by independent peer reviewers. The reviewers appreciated the attention to an important problem, but raised some substantial concerns about the current manuscript. Based on the reviews, we will not be able to accept this version of the manuscript, but we would be willing to review a much-revised version. We cannot, of course, promise publication at that time.

If you decide to revise the manuscript for further consideration at PLOS Genetics, please aim to resubmit within the next 60 days, unless it will take extra time to address the concerns of the reviewers, in which case we would appreciate an expected resubmission date by email to plosgenetics@plos.org.

If present, accompanying reviewer attachments are included with this email; please notify the journal office if any appear to be missing. They will also be available for download from the link below. You can use this link to log into the system when you are ready to submit a revised version, having first consulted our Submission Checklist .

PLOS has incorporated Similarity Check , powered by iThenticate, into its journal-wide submission system in order to screen submitted content for originality before publication. Each PLOS journal undertakes screening on a proportion of submitted articles. You will be contacted if needed following the screening process.

We are sorry that we cannot be more positive about your manuscript at this stage. Please do not hesitate to contact us if you have any concerns or questions.

Yours sincerely,

Gao Wang

Guest Editor

PLOS Genetics

Michael Epstein

Section Editor

PLOS Genetics

Reviewer's Responses to Questions

**Comments to the Authors:**

Reviewer #1: see attachment

Reviewer #2: This was an interesting manuscript that will be of interest/relevance to others working in the field. I have a few minor comments and two major comments.

Major Comments

-------------------

1. Much of the interest in fine-mapping relates to the situation where there is more than one causal variant in a region. Indeed the authors explore this a bit in their UKB analyses. In the simulations, it is unclear to me whether many/any of the regions that ended up being fine-mapped will have included more than one causal variant? Given the simulation parameters where \pi, the proportion of causal SNPs, is set to either 0.1% or 1%, it seems to me rather unlikely that any of the fine-mapped regions will have ended up harbouring more than one causal variant.

The authors need to include some simulation scenarios where they enforce a situation whereby there are multiple causal variants in a fine-mapped region. They then need took at the performance of the various methods *separately* in this situation (for comparison with the performance of the methods when there is only one causal variant per region).

2. The authors point out that they do not make any examination of the CSs that are actually output by FINEMAP, SuSIE-RSS and SuSIE-Inf - instead they define CSs using their own algorithm (essentially the same algorithm that they use for defining CSs from the BLR methods). Their justification for this is "the main objective of our study was to compare the efficiency of the algorithms of these models. Introducing a comparison based on the CSs they determine would introduce additional complexity and divert us far from our objective".

Unfortunately I do not consider this satisfactory. I consider it ESSENTIAL that the authors include in their comparison the CSs that are actually output by FINEMAP, SuSIE-RSS and SuSIE-Inf. The authors of these software packages (particularly for the SuSIE methods) worked very hard to come up with a principled way of defining CSs, indeed this was possibly the most attractive and compelling feature of the original SuSIE method when it was introduced. In any case this default option is what people will be using in practice. I have no objection to the authors additionally including their own definition CSs in the comparison, but the more interesting comparison is with the default outputs.

Minor Comments

-------------------

1. The Abstract goes into too much nitty gritty of detailed results (including percentages etc.) and as a result is quite confusing. It needs to be more succinct, with a focus on summarising the main findings.

2. Page 6 line 130: "Type two diabetes" should probably read "Type 2 diabetes".

3. Page 7 line 149: You mention UKB but have you correctly acknowledged UKB in the manuscript? At https://www.ukbiobank.ac.uk/enable-your-research/manage-your-project

It says:

Please ensure that you use the following in the Acknowledgments section in your paper:

"This research has been conducted using the UK Biobank Resource under Application Number xxxxx."

4. Page 9 line 192: Can you confirm that you will have ended up choosing different causal SNPs in each replicate of each simulation scenario?

5. Page 13 line 260: "Lloyd" not "Llyod". And this this presumably should be referred to as Reference 12?

6. Page 17 line 333-336: this slightly assumes that these regions of 500 SNPs to the left and right of each causal SNP actually came up as "significant "(i.e exceeding threshold \alpha). What did you do if (in a given simulation replicate) a region did not meet this criterion?

7. Page 21 lines 426-431: In Figure 4 you show results for "Power", not Recall. How do you define Power? Or are you using the terms "Power" and "Recall" interchangeably? (Page 40 lines 792-795 suggests this may be the case).

8. Page 28 lines 569-585: I do not find this a very interesting summary of the results. Can you work a bit harder to summarise the results in Fig 4? Which methods are "best"? Does it matter that (for example) bR3 and bR2 have high F1score but very low Precision?

9. Page 29 lines 587-595 (see also Page 39 lines 769-770). It is not really surprising that there are significant differences between the results for the different scenarios...

Reviewer #3: Thank you for the comprehensive manuscript and extensive comparisons you have performed.

Major comments:

1.) As you discussed in "Influence of parameters in simulations" it is difficult to disentangle under the setting of simulated parameters of each fine-mapped region. It would therefore be beneficial for comparisons of the methods if simulations parameters can set per genomic regions and xxx replicates are generated with the same parameters. A simple approach would be to use the list of identified T2D genes in Mahajan et al. Extract genotypes from UKB for a region around the genes and simulate phenotypes according to each combination of parameter values you want to evaluate.

2.) You are comparing fine-mapping methods under the assumption of one causal SNP per fine-mapping region. It is unclear why that assumption is necessary. In addition, it does not reflect the genomic architecture in T2D and other phenotypes (https://www.nature.com/articles/s41588-018-0241-6#Fig3). Therefore, it would be helpful for users of fine-mapping methods to see comparisons of the different methods when there is multiple causal signals presents in the each simulated region. This should be straightforward to implement in 1.) where you allow for xxx causal variants per simulated regions that jointly explain yyy% of phenotypic variance.

3.) Although you have made an effort in "Credible sets for simulations" to explain how credible sets are defined, I have problems understanding exactly how they are defined. Could you please explain for a single fine-mapped region, how a credible set is defined and how many credible sets you obtained per that one fine-mapped region?

4.) When comparing different models, it would also be beneficial for the community to see a comparison based on PIPs. That is, using ROC curves an AUCs as has been done is previous benchmarks of fine-mapping methods.

Minor comments:

1) Since you are allowing only for a single causal variant, it would make sense to run external fine-mapping methods such that they allow for exactly one causal variant.

2) It is unclear from the manuscript how PIPs are defined for BLR models.

3) It is not evident how you BLR methods would deal with genotype missingness and varying sample size per variant when applied for fine-mapping.

**Have all data underlying the figures and results presented in the manuscript been provided?**

Reviewer #1: Yes

Reviewer #2: Yes

Reviewer #3: Yes

PLOS authors have the option to publish the peer review history of their article (what does this mean? ). If published, this will include your full peer review and any attached files.

**Do you want your identity to be public for this peer review?** For information about this choice, including consent withdrawal, please see our Privacy Policy .

Reviewer #1: No

Reviewer #2: No

Reviewer #3: No

---

## [Decision Letter · Decision Letter 1]

PGENETICS-D-24-00378R1

Evaluation of Bayesian Linear Regression Models as a Fine Mapping tool

PLOS Genetics

Dear Dr. Shrestha,

Thank you for submitting your revised manuscript to PLOS Genetics. Reviewers were only partially satisfied with your responses; some important initial critiques were not addressed whereas some of the other new results led to serious confusion among the reviewers. Based on reviewer feedback, we are willing to provide you one additional chance to submit a revision that comprehensively addresses all reviewer concerns. Please note: if reviewers still feel the next revision has major criticisms, then we would in all likelihood reject the manuscript. 

Please submit your revised manuscript within 60 days Apr 26 2025 11:59PM. If you will need more time than this to complete your revisions, please reply to this message or contact the journal office at plosgenetics@plos.org. Please include the following items when submitting your revised manuscript:

We look forward to receiving your revised manuscript.

Kind regards,

Gao Wang

Guest Editor

PLOS Genetics

Michael Epstein

Section Editor

PLOS Genetics

Aimée Dudley

Editor-in-Chief

PLOS Genetics

Anne Goriely

Editor-in-Chief

PLOS Genetics

**Journal Requirements:**

At this stage, the following Authors/Authors require contributions: Merina Shrestha, Zhonghao Bai, Tahereh Gholipourshahraki, Astrid J. Hjelholt, Sile Hu, Mads Kjølby, Palle Duun Rohde, and Peter Sørensen. Please ensure that the full contributions of each author are acknowledged in the "Add/Edit/Remove Authors" section of our submission form.

The list of CRediT author contributions may be found here: https://journals.plos.org/plosgenetics/s/authorship#loc-author-contributions

2) Please update the Data Availability Statement in the online submission form to include the link to access the "The simulated phenotypes." 

**Reviewers' comments:**

Reviewer's Responses to Questions

Reviewer #1: Thank you to the authors for addressing most of my previous comments. The revised manuscript is more comprehensive; however, I still have the following concerns: 1. In this revision, the authors added a simulation comparison for two causal variants. In Table 4, they present a comparison of the credible set (CS) sizes, including the median size. However, what is LD between the two causal variants considered? The authors showed the new CS definition reduced the number of variants in each CS. This conclusion is somewhat surprising to me. Could the author further analyze the results and provide a more detailed comparison between the new and previous CS definitions. 2. Comment 1 also raises another question about the construction of CS. In the section "Credible sets for simulations with multiple causal SNPs", all variants are sorted by PIP and the CS are constructed based on the top sorted variants. At step 3, if a CS contains more than one SNP, the LD among these variants should be >0.5. In the additional simulations, if two causal variants have LD >0.5, the new CS will include both variants along with other highly correlated variants in the same CS at the construction part. It is unclear why the new CS construction results in a reduced number of SNPs. The authors should provide further clarification on this point.

Reviewer #2: This revised manuscript is improved by the addition of some simulations where the authors enforce a situation whereby there are multiple causal variants in a fine-mapped region. Unfortunately their description and discussion of the results obtained (lines 639-670 , page 30-31, S4 Table) seems to me quite opaque. The only thing that I find at all comprehensible are the AUCs mentioned in the text - but surely the full set of AUCs should be presented as a table (or supplementary table) and surely there should be some figures showing F1score, Precision and Recall? I do not understand the distinction the authors are making between CS>10 and CS <=10 and I really not understand what we are supposed to learn from the results shown in S4 Table and the text in lines 653-670. The authors need to do a much better job at guiding the reader through these results.

Apart from the simulations where there are multiple causal variants in a fine-mapped region, the authors seem to have chosen NOT to address my previous comment regarding including the "default" CSs that are actually output by FINEMAP, SuSIE-RSS and SuSIE-Inf. These are included in S4 Table (denoted by _a) but as far as I can tell they have NOT been included in any of the other tables or figures (e.g. S3 Table, Fig 4, S2 Fig, S3 Fig). I find this omission incomprehensible, seeing as I previously stated that I considered it ESSENTIAL that the authors include in their comparison the CSs that are actually output by FINEMAP, SuSIE-RSS and SuSIE-Inf, and, in their response, the authors stated "We fully agree that the default CS definitions are an integral aspect of these methods and a key feature that practitioners rely on."

Reviewer #3: Thank you for the comprehensive manuscript and extensive comparisons you have performed as well as addressing the reviewer comments.

Major:

You mention in the Author Summary that "that the BLR models better identify the genetic variants in terms of F1 score (recall and precision) and area under the receiver operating characteristic curve (AUC), in simulations." However, in the results section you state that "Highest average AUC estimate was observed for FINEMAP-Inf (0.80) followed by SuSIE-Inf (0.79), and 2 (0.79). Highest 1 . score, averaged across all the twenty-four simulation (binary and quantitative) scenarios was observed for the BayesR region-wide model ( 2 ) [ 1 . score: 0.40] followed by SuSIE-Inf [ 1 . score: 0.35] and SuSIE-RSS [ 1 . score: 0.34] (Fig 1)." In addition, you state in "Comparison of fine-mapping models in “Two causal genetic variants”" that "We observed higher AUC estimates for BayesR ( 1: 0.82, 2: 0.85) compared to BayesC ( 1 and 2: 0.76). Highest AUC estimate was observed for FINEMAP and FINEMAP-Inf (0.90) followed by SuSIE-Inf (0.87), BayesR and SuSIE-RSS (0.79)."

How are these results compatible with your statements in the Author summary?

Minor:

Would it be possible to address whether F1 score or AUC is more suitable for comparing methods?

The "Application in simulations" has become a bit unorganized and hard to follow. Would it be possible to make the results more concise in order to crystalize whether BLR methods improve fine-mapping over existing methods?

**Have all data underlying the figures and results presented in the manuscript been provided?**

Reviewer #1: Yes

Reviewer #2: Yes

Reviewer #3: Yes

PLOS authors have the option to publish the peer review history of their article (what does this mean? ). If published, this will include your full peer review and any attached files.

**Do you want your identity to be public for this peer review?** For information about this choice, including consent withdrawal, please see our Privacy Policy .

Reviewer #1: No

Reviewer #2: No

Reviewer #3: No

**Figure resubmission:**
---

## [Decision Letter · Decision Letter 2]

Dear Dr Sørensen,

We are pleased to inform you that your manuscript entitled "Enhanced Genetic Fine Mapping Accuracy with Bayesian Linear Regression Models in Diverse Genetic Architectures" has been editorially accepted for publication in PLOS Genetics. Congratulations!

Yours sincerely,

Gao Wang

Guest Editor

PLOS Genetics

Michael Epstein

Section Editor

PLOS Genetics

Aimée Dudley

Editor-in-Chief

PLOS Genetics

Anne Goriely

Editor-in-Chief

PLOS Genetics

Comments from the reviewers (if applicable):

Reviewer's Responses to Questions

**Comments to the Authors:**

Reviewer #1: The authors have addressed most of my comments. There is a minor concern about construction using thr ranked PIP and with the additional cutoff to the lead variant. If the lead variant is a spurious signal, it may cause the false discoveries; additionally, there may exist a risk that two variants have LD > 0.5 with the lead variant but LD < 0.5 between two variants, they will within the same CS, which is a key differnece between checking purity of entire CS and purity with lead variant. The authors could add a discussion in the manuscript.

Reviewer #2: My previous concerns have now been satisfactorily addressed.

**Have all data underlying the figures and results presented in the manuscript been provided?**

Reviewer #1: Yes

Reviewer #2: Yes

PLOS authors have the option to publish the peer review history of their article (what does this mean? ). If published, this will include your full peer review and any attached files.

**Do you want your identity to be public for this peer review?** For information about this choice, including consent withdrawal, please see our Privacy Policy .

Reviewer #1: No

Reviewer #2: No

**Data Deposition**

http://datadryad.org/submit?journalID=pgenetics&manu=PGENETICS-D-24-00378R2

**Press Queries**

---

## [Editor Report · Acceptance letter]

PGENETICS-D-24-00378R2

Enhanced Genetic Fine Mapping Accuracy with Bayesian Linear Regression Models in Diverse Genetic Architectures

Dear Dr Sørensen,

We are pleased to inform you that your manuscript entitled "Enhanced Genetic Fine Mapping Accuracy with Bayesian Linear Regression Models in Diverse Genetic Architectures" has been formally accepted for publication in PLOS Genetics! Your manuscript is now with our production department and you will be notified of the publication date in due course.

With kind regards,

Judit Kozma

PLOS Genetics

On behalf of:
